# QCD Theory of the Hadrons and Filling the Yang–Mills Mass Gap

**Jay R. Yablon** 

Einstein Centre for Local-Realistic Physics, 15 Thackley End, Oxford OX2 6LB, UK; yablon@alum.mit.edu

**Abstract:** The rank-3 antisymmetric tensors which are the magnetic monopoles of SU(N) Yang–Mills gauge theory dynamics, unlike their counterparts in Maxwell's U(1) electrodynamics, are non-vanishing, and do permit a net flux of Yang–Mills analogs to the magnetic field through closed spatial surfaces. When electric source currents of the same Yang–Mills dynamics are inverted and their fermions inserted into these Yang–Mills monopoles to create a system, this system in its unperturbed state contains exactly three fermions due to the monopole rank-3 and its three additive field strength gradient terms in covariant form. So to ensure that every fermion in this system occupies an exclusive quantum state, the Exclusion Principle is used to place each of the three fermions into the fundamental representation of the simple gauge group with an SU(3) symmetry. After the symmetry of the monopole is broken to make this system indivisible, the gauge bosons inside the monopole become massless, the SU(3) color symmetry of the fermions becomes exact, and a propagator is established for each fermion. The monopoles then have the same antisymmetric color singlet wavefunction as a baryon, and the field quanta of the magnetic fields fluxing through the monopole surface have the same symmetric color singlet wavefunction as a meson. Consequently, we are able to identify these fermions with colored quarks, the gauge bosons with gluons, the magnetic monopoles with baryons, and the fluxing entities with mesons, while establishing that the quarks and gluons remain confined and identifying the symmetry breaking with hadronization. Analytic tools developed along the way are then used to fill the Yang–Mills mass gap.

**Keywords:** hadrons; baryons; mesons; quarks; gluons; QCD; hadronization; quark-gluon plasma; Yang–Mills mass gap

## 1. Introduction

After the discovery of the muon in 1936, Rabi is said to have exclaimed: "who ordered that?" But to this day, the same question can still be asked of the proton and neutron which are at the nuclear heart of the observed material universe, and of the other baryons. We do have a very good understanding that the proton and neutron and other baryons are composed of three confined "colored" quarks in the fundamental representation of SU(3), with highly non-linear gluonic interactions among these quarks, wherein baryons interact with one another by exchanging a variety of mesons. But we still do not have a good dynamic answer, rooted in fundamental physics principles, to Rabi's very basic question: who ordered the baryons? Nor is there a good understanding of the dynamic origin of quark and gluon confinement.

One of the most notable features of Maxwell's U(1) differential and integral equations $\nabla \cdot \mathbf{B} = 0$ and $\oiint \mathbf{B} \cdot d\mathbf{S} = 0$ is that magnetic monopoles do not exist and that there is never a net flux of magnetic fields across any closed two-dimensional spatial surface. But in the dynamic Maxwell equations for an SU(N) Yang–Mills [1] theory of non-commuting gauge fields these monopoles do exist, and there is a net flux of the analogs to magnetic fields through closed surfaces surrounding and within the

monopole. Given the extraordinary success of Yang–Mills gauge theory in describing weak and strong interactions, and its presumably significant role in any "grand unified theory" which might become generally accepted in the future, the question how the magnetic monopoles of Yang–Mills might manifest themselves in the natural world must be given due consideration.

What is shown here is that when the Yang–Mills (YM) analog of Maxwell's electric charge equation is inverted, then inserted into the analog of Maxwell's magnetic equation for what are now non-vanishing monopoles—effectively combining both of Maxwell's covariant equations into a single indivisible equation—the non-perturbative state of these monopoles contains exactly 3 fermions arising from the monopole density being an antisymmetric tensor of rank 3. Treating the monopole as a "system" to which the Exclusion Principle must be applied, and so using SU(3) to enforce an exclusive quantum state for each of these three fermions, and following a form of spontaneous symmetry breaking which moves a degree of freedom from gauge bosons to fermions, makes the bosons massless and renders SU(3) an exact symmetry, these YM magnetic monopoles acquire the same SU(3) antisymmetric color singlet wavefunction as baryons, and the magnetic field analogs which flow through the monopole surfaces obtain the same symmetric color singlet wavefunction as mesons. This enables the three fermion states inside the monopole to be identified as confined colored quarks, the gauge bosons to be identified as confined colored gluons, the YM magnetic monopoles to be identified as baryons, the mesons to be identified as quanta of the YM magnetic fields which net flow in and out of these monopoles, and the symmetry breaking to be identified with ultra-high-energy hadronization from a plasma of free quarks and gluons.

From this we also answer Rabi's question: baryons were ordered by Maxwell, Yang and Mills, with an assist from Weyl via gauge theory itself, from Fermi–Dirac–Pauli via Dirac's quantum theory of the electron and the Exclusion Principle, from Gauss's law for fluxes of magnetic fields through closed surfaces, from Einstein's generally covariant formulation of magnetic monopoles as third-rank antisymmetric tensors, and with credit to Hamilton for pioneering non-commuting quaternions which later became the foundation of YM gauge theory in the form of Pauli matrices and their extension to SU(N).

Finally, we employ the foregoing development to fill the Yang–Mills Mass Gap [2].

## 2. A Brief Review of Maxwell's Equations Using Duality and Differential Forms

We start with Maxwell's equations in covariant form, in flat spacetime. A gauge field/vector potential $A^\nu$ with dimensionality of energy/charge is used to first define a field strength tensor $F^{\mu\nu} = \partial^\mu A^\nu - \partial^\nu A^\mu$ which in turn is used in the two differential Maxwell equations:

$$c\mu_0 j^\nu = \partial_\sigma F^{\sigma\nu} = \partial_\sigma \partial^\sigma A^\nu - \partial_\sigma \partial^\nu A^\sigma = (g^{\mu\nu}\partial_\sigma\partial^\sigma - \partial^\mu\partial^\nu)A_\mu \tag{1a}$$

$$c\mu_0 p^{\sigma\mu\nu} = \partial^\sigma F^{\mu\nu} + \partial^\mu F^{\nu\sigma} + \partial^\nu F^{\sigma\mu} = 0 \tag{1b}$$

Additionally, from (1a) we obtain the continuity equation:

$$c\mu_0 \partial_\nu j^\nu = \partial_\nu\partial_\sigma\partial^\sigma A^\nu - \partial_\nu\partial_\sigma\partial^\nu A^\sigma = 0 \tag{2}$$

Above, $j^\nu = (c\rho, \mathbf{j})$ is a current density four-vector in which $\rho$ has dimensions of charge per volume (charge density), $\mu_0$ is the vacuum permeability of free space, and $p^{\sigma\mu\nu}$ is a third-rank antisymmetric tensor defining a magnetic charge (monopole) current density.

In flat spacetime where $g_{\mu\nu} = \eta_{\mu\nu}$ and the Riemann tensor $R_{\mu\nu\alpha\sigma} = 0$, the commutator $[\partial_{;\mu}, \partial_{;\nu}]A_\alpha = R_{\mu\nu\alpha\sigma}A^\sigma$ for covariant derivatives simplifies to $[\partial_\mu, \partial_\nu] = 0$ with partial derivatives commuting. As a result, in (1a) we obtain the configuration space operator $g^{\mu\nu}\partial_\sigma\partial^\sigma - \partial^\mu\partial^\nu$ operating on $A_\mu$; and in (1b) we find the magnetic monopole density $p^{\sigma\mu\nu} = 0$, by identity, which is understood to mean there are no isolated magnetic charges in nature (setting aside the monopoles theorized as a possibility by Dirac in [3]). Likewise, (2) is also true by identity and governs the conservation of electric charge

sources. If we impose the covariant gauge condition $\partial_\sigma A^\sigma = 0$ then (1a) simplifies to $c\mu_0 j^\nu = \partial_\sigma \partial^\sigma A^\nu$. If, instead, we introduce a Proca mass $m \neq 0$ so (1a) becomes $c\mu_0 j^\nu = \left(g^{\mu\nu}\left(\partial_\sigma \partial^\sigma + m^2\right) - \partial^\mu \partial^\nu\right)A_\mu$, then because of (2) we find that $m^2 \partial_\sigma A^\sigma = 0$, thus $\partial_\sigma A^\sigma = 0$, which is no longer a gauge condition but a continuity requirement.

Duality and differential forms express the differential Maxwell's Equations (1) in a compact form which simplifies obtaining the integral Maxwell's equations. Using duality reviewed, e.g., at pp. 87–89 of [4], we may rewrite these as:

$$c\mu_0 * j^{\sigma\mu\nu} = \partial^\sigma * F^{\mu\nu} + \partial^\mu * F^{\nu\sigma} + \partial^\nu * F^{\sigma\mu} \tag{3a}$$

$$c\mu_0 p^{\sigma\mu\nu} = \partial^\sigma F^{\mu\nu} + \partial^\mu F^{\nu\sigma} + \partial^\nu F^{\sigma\nu} = 0 \tag{3b}$$

Then, via differential forms reviewed, e.g., in Chapter 4 of [4] and pp. 218–220 of [5], we compact:

$$c\mu_0 * j = d * F = d * dA \tag{4a}$$

$$c\mu_0 p = dF = ddA = 0 \tag{4b}$$

Likewise, the continuity Equation (2) becomes:

$$c\mu_0 d * j = dd * F = dd * dA = 0 \tag{5}$$

Both (4b) and (5) apply the differential forms relation $dd = 0$ that the exterior derivative of the exterior derivative is zero, see, e.g., Section 4.6 of [4]. (Above, we adopt the convention that the indexes of a dual object match those on the left side of the Levi-Civita tensor, so $*X^{\sigma\mu\nu} = \varepsilon^{\sigma\mu\nu\alpha} X_\alpha$, $*Y^{\mu\nu} = \frac{1}{2!}\varepsilon^{\mu\nu\alpha\beta} Y_{\alpha\beta}$ and $*Z^\nu = \frac{1}{3!}\varepsilon^{\nu\alpha\beta\gamma} Z_{\alpha\beta\gamma}$ for any vector $X_\alpha$ and antisymmetric tensors $Y_{\alpha\beta}$ and $Z_{\alpha\beta\gamma}$. This is to ensure that (3a) and (4a) maintain the same relative sign between $*j$ and $d * F$.)

The form relations (4) are ideal for casting Maxwell's equations into integral form. Specifically, for any $p$-form $H$ on a $p + 1$-dimensional manifold $M$ with boundary $\partial M$ and with $dH = (-1)^p \partial_\mu H dx^\mu$:

$$\int_M dH = \int_{\partial M} H \tag{6}$$

see, e.g., pp. 218–220 of [5]. So, using (4) in (6), we arrive at the integral Maxwell's equations:

$$c\mu_0 \iiint *j = \iiint d * F = \oiint *F = \oiint *dA \tag{7a}$$

$$c\mu_0 \iiint p = \iiint dF = \oiint F = \oiint dA = \iiint 0 = 0 \tag{7b}$$

In Section 18.3 of [6], Close uses Gauss's theorem for electric charge contained in (7a) to "consider the chromodynamics case which is analogous to the above." Similarly, (7b) contains Gauss's law for magnetism, $\oiint \mathbf{B} \cdot d\mathbf{S} = 0$. The reason these surface integrals are of interest is because one way to state the confinement of net color charge inside a baryon is via the schematic expression $\oiint_B$ net color $= 0$ over the baryon $B$ surface. Likewise, with only color-neutral objects (e.g., mesons) net flowing across baryon surfaces, we may write $\oiint_B$ color-neutral $\neq 0$. Because confinement is fundamentally about what can and cannot flow across baryon surfaces, Gauss's theorem will be of keen interest in the development to follow.

## 3. Maxwell's Yang–Mills Canonic Equations

Maxwell's electrodynamics is a U(1) abelian gauge theory, so named because its gauge fields are commuting, $\left[A_\mu, A_\nu\right] = 0$. Yang–Mills gauge theory, on the other hand, uses gauge fields we denote generally as $G^\nu$ to distinguish from $A^V$, which are non-commuting, $\left[G_\mu, G_\nu\right] \neq 0$. Specifically, for a

simple group SU(N) with traceless N×N Hermitian generators $\tau_i = \tau_i{}^\dagger$ with $i = 1 \ldots N^2 - 1$ normalized to $\mathrm{tr}(\tau_i^2) = \frac{1}{2}$ for each $\tau_i$, and a commutator $[\tau_i, \tau_j] = i f_{ijk} \tau_k$, these gauge fields are constructed via $G^\mu = \tau_i G_i^\mu = G^{\mu\dagger}$ and so are likewise N×N Hermitian matrices. We may use the commutator to find that $[G_\mu, G_\nu] = [\tau_i, \tau_j] G_{i\mu} G_{j\nu} = i f_{ijk} \tau_k G_{i\mu} G_{j\nu}$. Likewise, while $\mathrm{tr}(G^\mu) = \mathrm{tr}(\tau_i G_i^\mu) = 0$, it can be generally shown that $\mathrm{tr}(AB) = \mathrm{tr}(\tau_i \tau_j A_i B_j) = \frac{1}{2} A_i B_i$ for any $A = \tau_i A_i$, $B = \tau_j B_j$, which means that $\mathrm{tr}(G_\mu G_\nu) = \mathrm{tr}(\tau_i \tau_j G_{i\mu} G_{j\nu}) = \frac{1}{2} G_{i\mu} G_{i\nu}$ for a product of two gauge fields. Although weak interactions use SU(2) and strong interactions SU(3), at the outset we shall not examine any particular gauge groups. Our immediate interest is to develop the counterparts to Maxwell's equations generally, for any SU(N) Yang–Mills (YM) gauge theory.

In general, a gauge-covariant derivative is defined by $hcD^\mu = hc\partial^\mu - igG^\mu$, where $g$ is a charge strength. We then use these to define a field strength tensor in natural units $h = c = 1$ by:

$$F^{\mu\nu} = D^\mu G^\nu - D^\nu G^\mu = (\partial^\mu - igG^\mu)G^\nu - (\partial^\nu - igG^\nu)G^\mu = \partial^\mu G^\nu - \partial^\nu G^\mu - ig[G^\mu, G^\nu] \tag{8}$$

Were these gauge fields to commute, that is, were we to have $[G^\mu, G^\nu] = 0$, this would reduce to $F^{\mu\nu} = \partial^\mu G^\nu - \partial^\nu G^\mu$ which recovers the template $F^{\mu\nu} = \partial^\mu A^\nu - \partial^\nu A^\mu$ for U(1) abelian electrodynamics noted prior to (1). Of course, however, these $[G_\mu, G_\nu] = i f_{ijk} \tau_k G_{i\mu} G_{j\nu}$ are non-commuting. Using differential forms, the above compacts to:

$$F = dG - igG^2 \tag{9}$$

In Yang–Mills gauge theory we obtain dynamic equations starting with a canonic replacement $\partial_\mu \to D_\mu$ of ordinary with gauge-covariant derivatives. Likewise promoting electric and magnetic charge densities $j^\nu \mapsto J^\nu$ and $p^{\sigma\mu\nu} \mapsto P^{\sigma\mu\nu}$ to capitalization, Maxwell's (1) become: (Note that $c\mu_0$ remains the constant factor in (10) below as it was in (2) et seq. In U(1) electrodynamics $c\mu_0 = 4\pi h \alpha_e / e^2$ is the ratio between the running fine structure coupling $\alpha_e(\mu = 0) = 1/137.036\ldots$ and charge strength $e$. For SU(N) it remains the ratio $c\mu_0 = 4\pi h \alpha / g^2$ between dimensionless running couplings $\alpha$ and charge strengths $g$, generally.)

$$c\mu_0 J^\nu = D_\sigma F^{\sigma\nu} = \partial_\sigma F^{\sigma\nu} - igG_\sigma F^{\sigma\nu} = D_\sigma D^\sigma G^\nu - D_\sigma D^\nu G^\sigma = (g^{\mu\nu} D_\sigma D^\sigma - D^\mu D^\nu)G_\mu \tag{10a}$$

$$c\mu_0 P^{\sigma\mu\nu} = D^\sigma F^{\mu\nu} + D^\mu F^{\nu\sigma} + D^\nu F^{\sigma\mu} = (\partial^\sigma - igG^\sigma)F^{\mu\nu} + (\partial^\mu - igG^\mu)F^{\nu\sigma} + (\partial^\nu - igG^\nu)F^{\sigma\mu} = 0 \tag{10b}$$

Likewise, the continuity Equation (2) generalizes to:

$$\begin{aligned} c\mu_0 D_\nu J^\nu = c\mu_0(\partial_\nu - igG_\nu)J^\nu &= D_\nu D_\sigma F^{\sigma\nu} = D_\nu D_\sigma D^\sigma G^\nu - D_\nu D_\sigma D^\nu G^\sigma \\ &= \partial_\nu \partial_\sigma F^{\sigma\nu} - \Big(ig(G_\nu \partial_\sigma + \partial_\nu G_\sigma) + g^2 G_\nu G_\sigma\Big)F^{\sigma\nu} = (\partial_\nu \partial_\sigma - V_{\nu\sigma})F^{\sigma\nu} = 0 \end{aligned} \tag{11}$$

which in momentum space $i\partial_\mu \mapsto p_\mu$ becomes $(p_\nu + gG_\nu)J^\nu = 0$. Unlike (1b) and (2), the zeros above arise not from derivative commutation, but from the Jacobian identity $[D^\sigma, [D^\mu, D^\nu]] + [D^\mu, [D^\nu, D^\sigma]] + [D^\nu, [D^\sigma, D^\mu]] = 0$ and $[D_\nu, [D_\sigma, [D^\sigma, D^\nu]]] = 0$, in view of the further identities $-igF_{\mu\nu}\phi = [D_\mu, D_\nu]\phi = D_\mu(D_\nu\phi) - D_\nu(D_\mu\phi)$ and $[D_\sigma, F_{\mu\nu}]\phi = D_\sigma F_{\mu\nu}\phi$ operating on any field $\phi(t, \mathbf{x})$, see, e.g., [7], with careful attention given to the product rule. In the bottom line of (11), we define $V_{\nu\sigma} \equiv ig(G_\nu \partial_\sigma + \partial_\nu G_\sigma) + g^2 G_\nu G_\sigma$ to be a "perturbation tensor," so named because its trace $V = V^\sigma{}_\sigma = ig(G^\sigma \partial_\sigma + \partial^\sigma G_\sigma) + g^2 G^\sigma G_\sigma$ is the standard expression for the perturbation in the Klein–Gordon (relativistic Schrödinger) equation, and houses the difference $-V = D_\sigma D^\sigma - \partial_\sigma \partial^\sigma$ between ordinary and gauge-covariant Laplacians.

## 4. Maxwell's Yang–Mills Dynamic Equations

One of the major lessons of the General Theory of Relativity [8] which Hermann Weyl later adapted to gauge theory [9–11] is the canonic prescription of invariantly maintaining the original form of a field equation or Lagrangian density while merely replacing all ordinary derivatives

with suitable covariant derivatives. Then, by separating the original equation or Lagrangian and its ordinary derivatives from the new terms arising via covariant derivatives, we ascertain the physical, *dynamic* impacts of this prescription. For example, in General Relativity the covariant promotion of derivatives in the flat spacetime commutator $\left[\partial_\mu, \partial_\nu\right] = 0$ produces $\left[\partial_{;\mu}, \partial_{;\nu}\right]A_\alpha = R_{\mu\nu\alpha\sigma}A^\sigma$ with the Riemann tensor; and Newton's first law of motion written as $du^\alpha/d\tau = 0$ with a four velocity $u^\alpha = dx^\alpha/d\tau$ produces $Du^\alpha/D\tau = du^\beta/d\tau + \Gamma^\beta_{\mu\nu}u^\mu u^\nu = 0$, i.e., $du^\alpha/d\tau = -\Gamma^\alpha_{\mu\nu}u^\mu u^\nu$, which is the gravitational geodesic equation. And in gauge theory, the Klein–Gordon equation $(\partial_\sigma\partial^\sigma + m^\sigma)\phi = 0$ promotes to $(D_\sigma D^\sigma + m^\sigma)\phi = 0$, i.e., $(\partial_\sigma\partial^\sigma + m^\sigma)\phi = V\phi$ with the perturbation $V = ig(\partial_\sigma G^\sigma + G_\sigma\partial^\sigma) + g^2 G_\sigma G^\sigma$ noted after (11); while Dirac's equation $(i\gamma^\sigma\partial_\sigma - m)\psi = 0$ promotes to $(i\gamma^\sigma D_\sigma - m)\psi = 0$, i.e., $(i\gamma^\sigma\partial_\sigma - m)\psi = -\gamma^0 V_D\psi$ with a Dirac (D) perturbation $\gamma^0 V_D \equiv g\gamma^\sigma G_\sigma$. In all instances, this canonic prescription is to (1) replace ordinary with covariant derivatives, then (2) segregate the original equation with ordinary derivatives to see the dynamic impact, whereby what was originally a "zero" becomes a "non- zero." In Section 3 we took the first step of promoting ordinary derivatives in Maxwell's equations to covariant derivatives to obtain canonic equations. Now, we segregate the original Maxwell equations to see the dynamic Yang–Mills content, and what the zeros become as non-zeros.

To do so, we first define lowercase-denoted electric and magnetic charge densities by $c\mu_0 j^\nu \equiv \partial_\sigma F^{\sigma\nu}$ and $c\mu_0 p^{\sigma\mu\nu} \equiv \partial^\sigma F^{\mu\nu} + \partial^\mu F^{\nu\sigma} + \partial^\nu F^{\sigma\mu}$ exactly as in the Maxwell equations (1), using ordinary derivatives of the field strength $F^{\mu\nu}$, which is now the Yang–Mills (8). Then, we use $c\mu_0 J^\nu = D_\sigma F^{\sigma\nu}$ and $c\mu_0 P^{\sigma\mu\nu} = D^\sigma F^{\mu\nu} + D^\mu F^{\nu\sigma} + D^\nu F^{\sigma\mu} = 0$ in (10) which have exactly the same form but for $\partial \mapsto D$ and $j, p \mapsto J, P$, together with $D^\mu = \partial^\mu - igG^\mu$ in natural units, to separate the ordinary derivatives of $F^{\mu\nu}$, as such:

$$
\begin{aligned}
c\mu_0 j^\nu \equiv \partial_\sigma F^{\sigma\nu} &= \partial_\sigma(D^\sigma G^\nu - D^\nu G^\sigma) = (g^{\mu\nu}\partial_\sigma D^\sigma - \partial^\mu D^\nu)G_\mu \\
&= (g^{\mu\nu}(\partial_\sigma\partial^\sigma - ig\partial_\sigma G^\sigma) - (\partial^\mu\partial^\nu - ig\partial^\mu G^\nu))G_\mu
\end{aligned}
\tag{12a}
$$

$$
\begin{aligned}
c\mu_0 p^{\sigma\mu\nu} \equiv \partial^\sigma F^{\mu\nu} + \partial^\mu F^{\nu\sigma} + \partial^\nu F^{\sigma\mu} &= igG^\sigma F^{\mu\nu} + igG^\mu F^{\nu\sigma} + igG^\nu F^{\sigma\mu} \\
&= \partial^\sigma(D^\mu G^\nu - D^\nu G^\mu) + \partial^\mu(D^\nu G^\sigma - D^\sigma G^\nu) + \partial^\nu(D^\sigma G^\mu - D^\mu G^\sigma) \\
&= -ig(\partial^\sigma[G^\mu, G^\nu] + \partial^\mu[G^\nu, G^\sigma] + \partial^\nu[G^\sigma, G^\mu]) \neq 0
\end{aligned}
\tag{12b}
$$

Comparing (12) with (10) we see the foregoing implies definitions $c\mu_0 j^\nu \equiv c\mu_0 J^\nu + igG_\sigma F^{\sigma\nu}$ and $c\mu_0 p^{\sigma\mu\nu} \equiv c\mu_0 P^{\sigma\mu\nu} + igG^\sigma F^{\mu\nu} + igG^\mu F^{\nu\sigma} + igG^\nu F^{\sigma\mu}$ between lowercase and uppercase source densities, also mindful from (10b) that $P^{\sigma\mu\nu} = 0$. In (12b) the ordinary derivative commutator $\left[\partial_\mu, \partial_\nu\right] = 0$ cancels terms just as in Maxwell's monopole Equation (1b). What is very important, however, is that the Yang–Mills dynamic Equation (12b) contains a non-vanishing magnetic monopole density $p^{\sigma\mu\nu} \neq 0$, versus $p^{\sigma\mu\nu} = 0$ in (1b) for Maxwell's U(1) electrodynamics. The "zero," which here has turned into a "nonzero," is the magnetic monopole density $p^{\sigma\mu\nu}$.

Cast into differential forms, (12) may be compacted to (compare (4)):

$$
c\mu_0 * j = d * F = d * DG = d * \left(dG - igG^2\right) = d * dG - igd * G^2
\tag{13a}
$$

$$
c\mu_0 p = dF = dDG = d\left(dG - igG^2\right) = ddG - igdG^2 = -igdG^2 \neq 0
\tag{13b}
$$

These are Maxwell's dynamic equations in differential forms, for any Yang–Mills gauge group SU(N). Although we continue to apply the exterior calculus relation $dd = 0$ to remove $ddG = 0$, again, these magnetic monopoles do not zero out entirely. There remains a residual term $-igdG^2$ which arises directly out of the two-form $G^2 = \frac{1}{2}\left[G_\mu, G_\nu\right]dx^\mu dx^\mu \neq 0$, that is, directly from the non-commuting nature of Yang–Mills gauge theories.

Another reason it is important to segregate ordinary derivatives is because the $d$ in $dH$ in (6) remains an ordinary, geometric, calculus derivative, and does not change for Yang–Mills theory.

So, having placed (12) into the compact form (13) with segregated ordinary derivatives, we may use (6) to recast these into integral form:

$$c\mu_0 \iiint *j = \iiint d*F = \oiint *F = \iiint d*\left(dG - igG^2\right) = \oiint *dG - ig \oiint *G^2 \tag{14a}$$

$$c\mu_0 \iiint p = \iiint dF = \oiint F = -ig \iiint dG^2 = -ig \oiint G^2 \neq 0 \tag{14b}$$

These are the Yang–Mills counterparts to (7). The YM "electric" equation (14a) contains a new term $-ig \oiint *G^2$ which does not appear in (7a), again, because $G^2 \neq 0$. The YM magnetic equation written as $\oiint F = -ig \oiint G^2$ indicates something quite unique in contrast to Maxwell's electrodynamics: Whereas (aside perhaps from Dirac's [3]) there is no net flux of any U(1) magnetic fields through any closed two-dimensional surface, we learn from (14b) that Yang–Mills SU(N) "magnetic" fields can and do exhibit a net flux through such closed surfaces. This is because (13b) does provide Yang–Mills theory with non-vanishing magnetic monopoles.

Yang–Mills gauge theories have proved to be very successful for understanding the natural world. Weak interactions are correctly understood using SU(2), strong using SU(3), and electroweak using SU(2)xU(1). It is plausible that grand unification will eventually start with some larger SU(N) and spontaneously break symmetry in stages down to the phenomenological SU(3)xSU(2)xU(1). In short, we take Yang–Mills gauge theories very seriously for their ability to render what we observe in nature. Accordingly, if Yang–Mills theories also predict magnetic monopoles as in (12b) and (13b), and non-zero magnetic field surface fluxes as in (14b), we must ask equally serious questions about these monopoles. Most importantly: in the natural world, in what form do we observe these YM monopoles? Furthermore, what are the $G^2$ objects which in (14b) net flow across closed surfaces around and within the monopole?

As we shall see, these monopoles are observed as baryons and these $G^2$ objects which net flow across surfaces of these monopoles are observed as mesons. Together, they are the hadrons.

## 5. Populating Yang–Mills Magnetic Monopoles with Source Currents, by Inverting the Yang–Mills Electric Source Equation and Then Combining Both Maxwell Equations into One

It is common practice to start with an electric equation of the form (1a), or, presently, (12a) in which the source density $j^\nu$ is a function of the gauge field $A_\mu$ or $G_\mu$, then invert the configuration space operator to obtain the gauge field as a function of the source current. It is well-known, however, that the inverse of (1a) is infinite and cannot be obtained without removing some of the gauge freedom, typically through the gauge condition $\partial_\sigma A^\sigma = 0$. Alternatively, a finite inverse can be obtained with a Proca mass $m$ added by hand, whereby (1a) is written as $c\mu_0 j^\nu = \left(g^{\mu\nu}\left(\partial_\sigma \partial^\sigma + m^2\right) - \partial^\mu \partial^\nu\right)A_\mu$. Of course, a theory with a vector boson mass added by hand is no longer renormalizable, but it is also known how to cure this by revealing a mass through spontaneous symmetry breaking of the sort which underlies electroweak theory. With this in mind, we now add a mass by hand to (12a), and with $h = c = 1$ write:

$$\begin{aligned} c\mu_0 j^\nu = \partial_\sigma F^{\sigma\nu} &= \left(g^{\mu\nu}\left(\partial_\sigma D^\sigma + m^2\right) - \partial^\mu D^\nu\right)G_\mu \\ &= \left(g^{\mu\nu}\left(\left(\partial_\sigma \partial^\sigma + m^2\right) + g^2 G_\sigma G^\sigma\right) - \left(\partial^\mu \partial^\nu - ig\partial^\mu G^\nu\right)\right)G_\mu \end{aligned} \tag{15}$$

With this same $m$, (10a) becomes $c\mu_0 J^\nu = \left(g^{\mu\nu}\left(D_\sigma D^\sigma + m^2\right) - D^\mu D^\nu\right)G_\mu$. Then we find from (11) that $m^2 D_\sigma G^\sigma = 0$ is now a required covariant condition—not merely an optional gauge condition—through which the scalar degree of freedom is removed from $G^\sigma$. So, for a massive vector boson in Yang–Mills theory:

$$D_\sigma G^\sigma = 0 \text{ i.e. } \partial_\sigma G^\sigma = igG_\sigma G^\sigma \tag{16}$$

which we already used in going from (12a) to the bottom line of (15). It is interesting—and a precursor to solving the Yang–Mills Mass Gap problem [2] in Section 17—that this produces a correctly signed

term $+g^2 G_\sigma G^\sigma$ in the configuration space operator of (15), because this is the form in which the electroweak Lagrangian $\mathcal{L} = D^\sigma * \phi * D_\sigma \phi = \partial^\sigma \phi * \partial_\sigma \phi + g^2 G^\sigma G_\sigma \phi * \phi$ reveals renormalizable gauge boson masses following spontaneous symmetry breaking at the Fermi vacuum expectation value (vev).

From here the inverse calculation is straightforward: Via (15) we define a tensor $I_{\alpha v}$ by:

$$c\mu_0 I_{\alpha v} j^v = I_{\alpha v}\left(g^{\mu v}\left(\partial_\sigma D^\sigma + m^2\right) - \partial^\mu D^v\right)G_\mu \equiv \delta^\mu{}_\alpha G_\mu = G_\alpha \tag{17}$$

Because $I_{\alpha v}\left(g^{\mu v}\left(\partial_\sigma D^\sigma + m^2\right) - \partial^\mu D^v\right) \equiv \delta^\mu{}_\alpha$, we see that $I_{\alpha v}$ is the left-side inverse of the configuration space operator $g^{\mu v}\left(\partial_\sigma D^\sigma + m^2\right) - \partial^\mu D^v$. Further, because the inverse $M^{-1}$ of any $N$-dimensional square matrix $M$ must commute with $M$, i.e., $M^{-1}M = MM^{-1} = \text{Id}$ where Id is a like-dimensioned identity matrix, the full specification of this inverse for operation on either side is:

$$I_{\alpha v\text{LEFT}}\left(g^{\mu v}\left(\partial_\sigma D^\sigma + m^2\right) - \partial^\mu D^v\right) = \left(g^{\mu v}\left(\partial_\sigma D^\sigma + m^2\right) - \partial^\mu D^v\right)I_{\alpha v\text{RIGHT}} \equiv \delta^\mu{}_\alpha \tag{18}$$

with the further requirement that $I_{\alpha v\text{LEFT}} = I_{\alpha v\text{RIGHT}} \equiv I_{\alpha v}$. Then, once we have an $I_{\alpha v}$ which satisfies (18), we may insert (17) written as $G^\mu = c\mu_0 I^{\mu\sigma} j_\sigma$ into (12b) with suitable index renaming, and use $c^2\mu_0\varepsilon_0 = 1$ where $\varepsilon_0$ is the free space vacuum permittivity, to obtain:

$$ic\varepsilon_0 p^{\alpha\mu v} = g\left(\partial^\alpha\left[I^{\mu\sigma} j_\sigma, I^{v\tau} j_\tau\right] + \partial^\mu\left[I^{v\tau} j_\tau, I^{\alpha v} j_\gamma\right] + \partial^v\left[I^{\alpha v} j_\gamma, I^{\mu\sigma} j_\sigma\right]\right) \tag{19}$$

This is important for two reasons: First, the U(1) Maxwell Equations (1) are two distinct equations because there are no magnetic monopoles. But in (19)—courtesy of the non-vanishing YM monopole (12b) and the YM current sources (12a) with Proca mass in (15), the latter inverted using (17)—the resulting YM monopole (19) combines both Yang–Mills Maxwell equations into a single equation. Second, this YM monopole $p^{\alpha\mu v}$ has now been populated with a triplet of YM source currents $j_\sigma$, $j_\tau$, $j_\gamma$ having three distinct indexes. So when expanded with $j^v = \tau_i j_i{}^v$, and with each source being related to fermion wavefunctions courtesy of Dirac's [12], the net result of (19), as we shall see, is that we have populated the Yang–Mills monopole with a triplet of fermions. Via the exclusion principle following symmetry breaking these will end up in the fundamental representation of an exact SU(3) group, and thus have the same character as a quark color triplet. This is the route to discovering that these YM monopoles possess all the Quantum Chromodynamics (QCD) properties of baryons.

## 6. Nonlinear Recursive Interactions Contained in the Inverse Yang–Mills Electric Equation

The next step is to explicitly calculate the inverse using (18), then insert this in (17) and (19). This inverse calculation is carried out in detail in Appendix A. The result including the $+i\varepsilon$ prescription is (A13). Before proceeding, however, let us establish notation conventions for representing the energy momentum of a particle four-vector: Whenever the energy-momentum is that of a fermion we shall use the notation $p^\mu$. For a massive vector boson we use $k^\mu$. And for a massless boson such as a photon or gluon we use $q^\mu$. Accordingly, with $c$ restored, because (A13) is for a massive vector boson, with a parenthetical $(G_\alpha \dots)$ for highlighting reasons to be momentarily reviewed, and using the quoted "denominator" from (A13), whereby we represent an inverse by $M^{-1} \equiv 1/''M''$ with that inverse placed at the spot denoted by a subscripted $\vee$, we shall write this result in (A13) as:

$$I_{\alpha v} = \frac{\vee\left(-g_{\alpha v} + \frac{k_v k_\alpha + g k_v(G_\alpha\dots)/c}{m^2 c^2}\right)}{''k_\sigma k^\sigma - m^2 c^2 - g^2(G_\sigma\dots)(G^\sigma\dots)/c^2 + i\varepsilon''} \tag{20}$$

Next, we use (20) in (17) to write:

$$G_\alpha(j^v, G_\alpha) = c\mu_0 I_{\alpha v} j^v = c\mu_0 \frac{\vee\left(-g_{\alpha v} + \frac{k_v k_\alpha + g k_v(G_\alpha\dots)/c}{m^2 c^2}\right)}{''k_\sigma k^\sigma - m^2 c^2 - g^2(G_\sigma\dots)(G^\sigma\dots)/c^2 + i\varepsilon''} j^v \tag{21}$$

When we contract the above from the left with another $j^\alpha$ to form the Lagrangian density term $j^\alpha G_\alpha$, the result represents the propagator for a YM vector boson mediating between two source currents, which interactions are routinely represented with Feynman diagrams.

Now, were it not for the Yang–Mills terms with $(G_\alpha \ldots)$ parenthetically highlighted in the above, we could say (21) inverts (16) to obtain $G_\alpha$ as a function $G_\alpha(j^\nu)$ exclusively of $j^\nu$. However, $G^\sigma$ *is* in (21), so that is *not* what (21) does. Rather, in (21) $G_\alpha(j^\nu, G_\alpha)$ is a recursive function of $j^\nu$ *and of itself*. For example, at the first recursion we substitute $G_\alpha$ on the left into the $(G_\alpha \ldots)$ to obtain:

$$
G_\alpha = c\mu_0 \frac{\vee\left(-g_{\alpha\nu} + \dfrac{k_\nu k_\alpha + gk_\nu \left(c\mu_0 \dfrac{\vee\left(-g_{\alpha\beta} + \frac{k_\beta k_\alpha + gk_\beta (G_\alpha \ldots)/c}{m^2c^2}\right)}{''k_\sigma k^\sigma - m^2c^2 - g^2(G_\sigma \ldots)(G^\sigma \ldots)/c^2 + i\varepsilon''} j^\beta\right)/c}{m^2c^2}\right)}{''k_\sigma k^\sigma - m^2c^2 - g^2\left(c\mu_0 \dfrac{\vee\left(-g_{\sigma\tau} + \frac{k_\tau k_\sigma + gk_\tau (G_\alpha \ldots)/c}{m^2c^2}\right)}{''k_\sigma k^\sigma - m^2c^2 - g^2(G_\sigma \ldots)(G^\sigma \ldots)/c^2 + i\varepsilon''} j^\tau\right)^2 /c^2 + i\varepsilon''} j^\nu
\tag{22}
$$

In the bottom denominator we use the shorthand $G_\sigma G^\sigma \equiv G^{\sigma 2}$. This type of substitution can and must be done indefinitely, approaching an infinite number of substitutions, before we truly have $G_\alpha(j^\nu)$ and not $G_\alpha(j^\nu, G_\alpha)$. We may symbolically represent this infinite recursive series by $G_\alpha(j^\nu, G_\alpha(j^\nu, G_\alpha(j^\nu, G_\alpha(j^\nu, G_\alpha(\ldots)))))$, which we condense into the notation $(G_\alpha \ldots)$ in (20) to (22). Obviously, the Lagrangian term $j^\alpha G_\alpha$ has a similarly recursive character.

Likewise, this recursion introduces an infinite number of occurrences of $j^\nu$ into (22), amplifying the nonlinearity of interactions amongst the $j^\nu$ themselves. This should not be surprising, because it is well-known Yang–Mills gauge theories are highly non-linear: Using (8), the Lagrangian density $\mathcal{L} = -\frac{1}{4}\partial^{[\mu}G_i^{\nu]}\partial_{[\mu}G_{i\nu]} - \frac{1}{2}gf_{ijk}\partial^{[\mu}G_i^{\nu]}G_{j\mu}G_{k\nu} - \frac{1}{4}g^2 f_{ijk}f_{ilm}G_j{}^\mu G_k{}^\nu G_{l\mu}G_{m\nu}$ in which $G^\mu = \tau_i G_i^\mu$ and $[\tau_i, \tau_j] = if_{ijk}\tau_k$, provides the basis for three- and four-gluon vertices. This is often used to illustrate the inherent non-linearity of particle interactions in Yang–Mills gauge theory, versus linear behaviors in QED where photons do not interact among themselves. So, the recursion in (22) is just another manifestation of this Yang–Mills non-linearity.

## 7. Introducing the Inverse Yang–Mills Electric Source Equation into the Yang–Mills Magnetic Monopoles, then Populating These Monopoles with Dirac Fermions

Next, we substitute the inverse (20) with renamed indexes as needed and recursion-highlighting parentheses removed, into the monopoles (19), to obtain:

$$
ic\varepsilon_0 p^{\alpha\mu\nu} = g
\begin{pmatrix}
\partial^\alpha \left[ \dfrac{\vee\left(-g^{\mu\sigma} + \frac{k^\sigma k^\mu + gk^\sigma G^\mu/c}{m^2c^2}\right)}{''k_\sigma k^\sigma - m^2c^2 - g^2 G_\sigma G^\sigma/c^2 + i\varepsilon''} j_\sigma, \dfrac{\vee\left(-g^{\nu\tau} + \frac{k^\tau k^\nu + gk^\tau G^\nu/c}{m^2c^2}\right)}{''k_\sigma k^\sigma - m^2c^2 - g^2 G_\sigma G^\sigma/c^2 + i\varepsilon''} j_\tau \right] \\[2em]
+\partial^\mu \left[ \dfrac{\vee\left(-g^{\nu\tau} + \frac{k^\tau k^\nu + gk^\tau G^\nu/c}{m^2c^2}\right)}{''k_\sigma k^\sigma - m^2c^2 - g^2 G_\sigma G^\sigma/c^2 + i\varepsilon''} j_\tau, \dfrac{\vee\left(-g^{\alpha\gamma} + \frac{k^\gamma k^\alpha + gk^\gamma G^\alpha/c}{m^2c^2}\right)}{''k_\sigma k^\sigma - m^2c^2 - g^2 G_\sigma G^\sigma/c^2 + i\varepsilon''} j_\gamma \right] \\[2em]
+\partial^\nu \left[ \dfrac{\vee\left(-g^{\alpha\gamma} + \frac{k^\gamma k^\alpha + gk^\gamma G^\alpha/c}{m^2c^2}\right)}{''k_\sigma k^\sigma - m^2c^2 - g^2 G_\sigma G^\sigma/c^2 + i\varepsilon''} j_\gamma, \dfrac{\vee\left(-g^{\mu\sigma} + \frac{k^\sigma k^\mu + gk^\sigma G^\mu/c}{m^2c^2}\right)}{''k_\sigma k^\sigma - m^2c^2 - g^2 G_\sigma G^\sigma/c^2 + i\varepsilon''} j_\sigma \right]
\end{pmatrix}
\tag{23}
$$

This provides a complete, explicit description of the way in which the YM monopoles are populated with the three source currents $j_\sigma$, $j_\tau$, $j_\gamma$, keeping in mind that every $G^\mu$ needs to be filled with an unlimited-approaching-infinite number of recursions as in (22).

When we further expand using $j^\nu = g\overline{\psi}\gamma^\nu\psi + \kappa^\nu$ obtained from the Yang–Mills continuity equation as reviewed in (A17) of Appendix B, we can directly populate (23) with Dirac fermions, as such:

$$ic\varepsilon_0 p^{\alpha\mu\nu} =$$

$$g \left[ \begin{array}{l} \partial^\alpha \left[ \dfrac{\sqrt{\left(-g^{\mu\sigma} + \frac{k^\sigma k^\mu + g k^\sigma G^\mu/c}{m^2 c^2}\right)}}{''k_\sigma k^\sigma - m^2 c^2 - g^2 G_\sigma G^\sigma/c^2 + i\varepsilon''} \left( \begin{array}{c} g\overline{\psi}\gamma_\sigma\psi \\ +\kappa_\sigma \end{array} \right) , \dfrac{\sqrt{\left(-g^{\nu\tau} + \frac{k^\tau k^\nu + g k^\tau G^\nu/c}{m^2 c^2}\right)}}{''k_\sigma k^\sigma - m^2 c^2 - g^2 G_\sigma G^\sigma/c^2 + i\varepsilon''} \left( \begin{array}{c} g\overline{\psi}\gamma_\tau\psi \\ +\kappa_\tau \end{array} \right) \right] \\ +\partial^\mu \left[ \dfrac{\sqrt{\left(-g^{\nu\tau} + \frac{k^\tau k^\nu + g k^\tau G^\nu/c}{m^2 c^2}\right)}}{''k_\sigma k^\sigma - m^2 c^2 - g^2 G_\sigma G^\sigma/c^2 + i\varepsilon''} \left( \begin{array}{c} g\overline{\psi}\gamma_\tau\psi \\ +\kappa_\tau \end{array} \right) , \dfrac{\sqrt{\left(-g^{\alpha\gamma} + \frac{k^\gamma k^\alpha + g k^\gamma G^\alpha/c}{m^2 c^2}\right)}}{''k_\sigma k^\sigma - m^2 c^2 - g^2 G_\sigma G^\sigma/c^2 + i\varepsilon''} \left( \begin{array}{c} g\overline{\psi}\gamma_\gamma\psi \\ +\kappa_\gamma \end{array} \right) \right] \\ +\partial^\nu \left[ \dfrac{\sqrt{\left(-g^{\alpha\gamma} + \frac{k^\gamma k^\alpha + g k^\gamma G^\alpha/c}{m^2 c^2}\right)}}{''k_\sigma k^\sigma - m^2 c^2 - g^2 G_\sigma G^\sigma/c^2 + i\varepsilon''} \left( \begin{array}{c} g\overline{\psi}\gamma_\gamma\psi \\ +\kappa_\gamma \end{array} \right) , \dfrac{\sqrt{\left(-g^{\mu\sigma} + \frac{k^\sigma k^\mu + g k^\sigma G^\mu/c}{m^2 c^2}\right)}}{''k_\sigma k^\sigma - m^2 c^2 - g^2 G_\sigma G^\sigma/c^2 + i\varepsilon''} \left( \begin{array}{c} g\overline{\psi}\gamma_\sigma\psi \\ +\kappa_\sigma \end{array} \right) \right] \end{array} \right] \tag{24}$$

Then, by an indefinite-number-approaching-infinity of recursions as reviewed at (21) and (22), the fermion and boson interactions inside these monopoles are seen to be highly nonlinear to infinite order. The final step is to show that (24) indeed represents the QCD properties of a baryon with three quarks and highly nonlinear gluon interactions among these quarks.

## 8. The Yang–Mills "Signal" Magnetic Monopole, without Perturbative "Noise"

It is well established that protons and neutrons, which are the two most important baryons insofar as they form the nuclei of the observed material universe, are teeming with non-linear interactions. For example, 2019 Particle Data Group (PDG) data [13] informs us that the free proton and neutron rest masses are $M_p = 938.272081 \pm 0.000006$ MeV and $M_n = 939.565413 \pm 0.000006$ MeV, but that the up and down current quark masses in an $\overline{\text{MS}}$ renormalization scheme at a scale $\mu \approx 2\,GeV$ are $m_u = 2.16^{+0.49}_{-0.26}$ MeV and $m_d = 4.67^{+0.48}_{-0.17}$ MeV, all respectively. With quark content p(duu) and n(udd), these "current quark" masses contribute about 1% to the overall rest energies of these baryons, with the other 99% arising from nonlinear gluon-mediated interactions among these quarks and from internal kinetic energies. These are distinguished from "constituent quark" masses which stem from attributing about a third of the total rest energy of a nucleon ($\approx$313 GeV) arising from their quark and gluon energies, to each quark. So, for a baryon, we may similarly coin "current baryon" to refer to the bare quark structure of the baryon with all nonlinearity stripped away, and "constituent baryon" to refer to the baryon including all its nonlinear behaviors. Here, going forward, we shall borrow the electrical engineering terms "signal" and "noise," and use the term "signal baryon" to refer to a baryon with all non-linear behaviors stripped away (the "current baryon"), and use "signal-plus-noise baryon" to refer to the entire observed baryon with all of its nonlinearity (the "constituent baryon"). Using this language, with each $G^\mu$ in (24) treated recursively in the manner of (22), what we have in (24) is clearly a "signal-plus-noise" monopole density. To explore the underlying QCD behaviors of these monopoles, we now shall study just the "signal" monopole density with all "noise" (which comprises about 99% of the observed rest energy of the proton and neutron and a large share of the rest energy of other baryons as well), removed.

It is fair to say that Yang–Mills gauge theory is a theory of perturbations added to Maxwell's linear electrodynamics in Section 2. For example, in the opening paragraph of Section 4 we find this in $V = ig(G^\sigma\partial_\sigma + \partial^\sigma G_\sigma) + g^2 G^\sigma G_\sigma = \partial_\sigma\partial^\sigma - D_\sigma D^\sigma$ arising from the second-order structure of the Klein–Gordon equation, and in $V_D = g\gamma^0\gamma^\sigma G_\sigma$ arising from the first-order Dirac equation. This is the "noise" of Yang–Mills theory. This is what Jaffe and Witten in [2] refer to as "excitations of the vacuum." If we wish to study just the "signal" without "excitations," the way to do so is to set the perturbations to zero and see what is left. So we do just that: Working from (23), the sources $j^\mu$ can be related to Dirac fermions via $j^\mu = g\overline{\psi}\gamma^\mu\psi + \kappa^\mu$ obtained in (A17). Then we set $V_D = 0$ which likewise means we have set $G_\sigma(t, \mathbf{x}) = 0$, thus $V_{\nu\sigma} = 0$. This removes from (23), all recursive non-linearity reviewed at (22), and it turns "denominators" into regular denominators without quotes. Still remaining are objects of the form $k^\sigma j_\sigma$. However, as seen following (A17), when $V_{\nu\sigma} = 0$ the continuity equation

is $p_\nu j^\nu = 0$ which in notation reviewed at the start of Section 6 means that $k^\sigma j_\sigma = 0$. So, all terms of this form as well as $k^\sigma k^\mu j_\sigma / m^2 c^2 = 0$ can be removed from (23). At this point all that remains are six numerator terms for which indexes can be raised via $-g^{\mu\sigma} j_\sigma = -j^\mu$, and signs cancelled. We finally segregate commutators in the numerator, so the pure "signal monopole" density inside the "noisy" (23) simply becomes:

$$ic\varepsilon_0 p^{\sigma\mu\nu} = g\frac{\partial^\sigma [j^\mu, j^\nu] + \partial^\mu [j^\nu, j^\sigma] + \partial^\nu [j^\sigma, j^\mu]}{(k_\tau k^\tau - m^2 c^2 + i\varepsilon)^2} \tag{25}$$

It is helpful to contrast (25) with (12b) from which it originated: All that has changed is that the gauge boson commutators $[G^\mu, G^\nu]$ have been replaced by the electric source commutators $[j^\mu, j^\nu]$, with the newly appearing denominators reflecting the inversion of Maxwell's Yang–Mills electric Equation (15) with mass $m$ into $G_\alpha(j^\nu)$ for the non-recursive signal monopole. Because the monopole is itself an N×N matrix for SU(N), with generators each normalized to $\text{tr}(\tau_i^2) = \frac{1}{2}$ we may additionally use the relation $\text{tr}(AB) = \text{tr}(\tau_i \tau_j A_i B_j) = \frac{1}{2} A_i B_i$ for $A = \tau_i A_i$ and $B = \tau_i B_i$, thus $\text{tr}[j^\mu, j^\nu] = \frac{1}{2}[j_i^\mu, j_i^\nu]$, to take the trace of both sides of (25), with the result that:

$$ic\varepsilon_0 \text{tr}\, p^{\sigma\mu\nu} = \frac{1}{2}g\frac{\partial^\sigma [j_i^\mu, j_i^\nu] + \partial^\mu [j_i^\nu, j_i^\sigma] + \partial^\nu [j_i^\sigma, j_i^\mu]}{(k_\tau k^\tau - m^2 c^2 + i\varepsilon)^2} \tag{26}$$

## 9. Populating the Yang–Mills "Signal" Magnetic Monopole with Dirac Fermions: Two Alternatives, Each of Which Shows That the Signal Monopole Contains Exactly Three Fermions

The next step is to populate the "signal" monopole trace (26) with Dirac fermions, similarly to what we did going from (23) to (24). The general relation $j^\nu = g\overline{\psi}\gamma^\nu \psi + \kappa^\nu$ between each $j^\nu$ and its fermion wavefunctions $\psi$ is (A17). However, since (25) is a signal monopole in which we have set $V_{\nu\sigma} = 0$ thus $\kappa^\nu = 0$, (A17) becomes $j^\nu = g\overline{\psi}\gamma^\nu \psi$. Because $j^\nu$ is an N×N matrix, the expansion of this is $j^\nu = \tau_i j_i^\nu = g\tau_i \overline{\psi}\tau_i \gamma^\nu \psi$ with $j_i^\nu = g\overline{\psi}\tau_i \gamma^\nu \psi$. For SU(N) these $\psi$ are Nx4 column vectors, with N arising from the Yang–Mills and 4 from the Dirac structure of each fermion. Because these sit in the fundamental representation of SU(N) we need to have N distinct SU(N) state labels for each $\psi$ and adjoint $\overline{\psi}$. There are two logical possibilities:

First, because $j_i^\mu = g\overline{\psi}\tau_i \gamma^\mu \psi$, for example, has a $\mu$ index, we may assign the label $\mu$ to this fermion and its adjoint and write $j_i^\mu = g\overline{\psi}_{(\mu)}\tau_i \gamma^\mu \psi_{(\mu)}$. Likewise for $j_i^\nu$ and $j_i^\sigma$. We may also label $m_{(\mu)}$ and $\varepsilon_{(\mu)}$ and $k_\mu k^\mu$ (the last with indexes doubling as labels) in the denominator, ditto for $\nu$ and $\sigma$. Doing this, and expanding commutators in (26), we obtain:

$$ic\varepsilon_0 \text{tr}\, p^{\sigma\mu\nu} = \frac{1}{2}g^3 \left( \begin{array}{l} \partial^\sigma \frac{\overline{\psi}_{(\mu)}\tau_i\gamma^\mu\psi_{(\mu)}\overline{\psi}_{(\nu)}\tau_i\gamma^\nu\psi_{(\nu)} - \overline{\psi}_{(\nu)}\tau_i\gamma^\nu\psi_{(\nu)}\overline{\psi}_{(\mu)}\tau_i\gamma^\mu\psi_{(\mu)}}{(k_\mu k^\mu - m_{(\mu)}^2 c^2 + i\varepsilon_{(\mu)})(k_\nu k^\nu - m_{(\nu)}^2 c^2 + i\varepsilon_{(\nu)})} \\ + \partial^\mu \frac{\overline{\psi}_{(\nu)}\tau_i\gamma^\nu\psi_{(\nu)}\overline{\psi}_{(\sigma)}\tau_i\gamma^\sigma\psi_{(\sigma)} - \overline{\psi}_{(\sigma)}\tau_i\gamma^\sigma\psi_{(\sigma)}\overline{\psi}_{(\nu)}\tau_i\gamma^\nu\psi_{(\nu)}}{(k_\nu k^\nu - m_{(\nu)}^2 c^2 + i\varepsilon_{(\nu)})(k_\sigma k^\sigma - m_{(\sigma)}^2 c^2 + i\varepsilon_{(\sigma)})} \\ + \partial^\nu \frac{\overline{\psi}_{(\sigma)}\tau_i\gamma^\sigma\psi_{(\sigma)}\overline{\psi}_{(\mu)}\tau_i\gamma^\mu\psi_{(\mu)} - \overline{\psi}_{(\mu)}\tau_i\gamma^\mu\psi_{(\mu)}\overline{\psi}_{(\sigma)}\tau_i\gamma^\sigma\psi_{(\sigma)}}{(k_\sigma k^\sigma - m_{(\sigma)}^2 c^2 + i\varepsilon_{(\sigma)})(k_\mu k^\mu - m_{(\mu)}^2 c^2 + i\varepsilon_{(\mu)})} \end{array} \right) \tag{27}$$

These fermions as well as objects in the denominators are now labelled with the index of the $j_i^\mu$, $j_i^\nu$ or $j_i^\sigma$ which contained them when we inserted fermions via source currents at (24), prior to removing the "noise."

Second, alternatively, although the fermions were introduced into the signal-plus-noise monopole at (24), once introduced, they have become part and parcel of a monopole system merging both Yang–Mills–Maxwell dynamic Equations (12), with (12a) given mass at (15) then inverted at (21). So, once this monopole system is established, we can change the labeling in (27) so each fermion and related denominator objects are labelled, not by the index of the source which brought them into the

monopole, but the index of the partial derivative acting on the fermion once it is in the monopole system, thus rendering the system indivisible. In this alternative (26) becomes:

$$
ic\varepsilon_0 \text{tr}\, p^{\sigma\mu\nu} = \frac{1}{2}g^3 \begin{pmatrix} \partial^\sigma \dfrac{\overline{\psi}_{(\sigma)}\tau_i\gamma^\mu\psi_{(\sigma)}\overline{\psi}_{(\sigma)}\tau_i\gamma^\nu\psi_{(\sigma)} - \overline{\psi}_{(\sigma)}\tau_i\gamma^\nu\psi_{(\sigma)}\overline{\psi}_{(\sigma)}\tau_i\gamma^\mu\psi_{(\sigma)}}{\left(k_\sigma k^\sigma - m_{(\sigma)}{}^2 c^2 + i\varepsilon_{(\sigma)}\right)^2} \\[2mm] + \partial^\mu \dfrac{\overline{\psi}_{(\mu)}\tau_i\gamma^\nu\psi_{(\mu)}\overline{\psi}_{(\mu)}\tau_i\gamma^\sigma\psi_{(\mu)} - \overline{\psi}_{(\mu)}\tau_i\gamma^\sigma\psi_{(\mu)}\overline{\psi}_{(\mu)}\tau_i\gamma^\nu\psi_{(\mu)}}{\left(k_\mu k^\mu - m_{(\mu)}{}^2 c^2 + i\varepsilon_{(\mu)}\right)^2} \\[2mm] + \partial^\nu \dfrac{\overline{\psi}_{(\nu)}\tau_i\gamma^\sigma\psi_{(\nu)}\overline{\psi}_{(\nu)}\tau_i\gamma^\mu\psi_{(\nu)} - \overline{\psi}_{(\nu)}\tau_i\gamma^\mu\psi_{(\nu)}\overline{\psi}_{(\nu)}\tau_i\gamma^\sigma\psi_{(\nu)}}{\left(k_\nu k^\nu - m_{(\nu)}{}^2 c^2 + i\varepsilon_{(\nu)}\right)^2} \end{pmatrix}
\tag{28}
$$

As we shall see, (27) and (28) are related through a form of symmetry breaking analogous in some ways to what is used in electroweak theory, which will become identified with hadronization.

Most important—in both (27) and (28)—is that this signal baryon is now seen to contain exactly *three fermions* $\psi_{(\sigma)}$, $\psi_{(\mu)}$, $\psi_{(\nu)}$ which arise from the rank 3 antisymmetric tensor which is the monopole $c\mu_0 p^{\sigma\mu\nu} = \partial^\sigma F^{\mu\nu} + \partial^\mu F^{\nu\sigma} + \partial^\nu F^{\sigma\mu}$ of (12b) and its three additive terms. This "three-ness" is structurally fundamental to the covariant representation of Maxwell's magnetic equations whether in U(1) Maxwell or in SU(N) Yang–Mills gauge theory. Consequently, the non-perturbative signal monopoles (27), (28) each describe a system—and (28) an indivisible system—built upon three distinct fermions. Just like a baryon.

## 10. Using the Gauge Group SU(3) to Establish Three Distinct Quantum States for the Three Fermions Populating a Yang–Mills Magnetic Monopole

The Fermi–Dirac–Pauli Exclusion Principle mandates that the fermions contained in any system of more than one fermion, e.g., an atom, nucleus, nucleon or baryon, must be distinguishable from all other fermions in that system by assignment of an exclusive quantum state. Accordingly, each of the fermions in (27) and (28)—being part of the monopole system—must have an exclusive quantum state. Because both (27) and (28) contain three fermions, the YM SU(N) gauge group used to provide this exclusion must have $N \geq 3$, so it cannot be SU(2). On the other hand, because there are exactly three fermions in both (27) and (28), there is no need for $N > 3$. Accordingly, we now use the group SU(3) to enforce Exclusion on the three fermions $\psi_{(\sigma)}$, $\psi_{(\mu)}$, $\psi_{(\nu)}$ in these alternative signal monopole systems (27) and (28). (We note without further detail here, that for these monopoles to be topologically stable, we must eventually employ SU(3) × U(1) following spontaneous symmetry breaking from a larger group, see Cheng and Li [14] at 472–473 and Weinberg [15] at 442. This U(1) generator provides the foundation for introducing hadron flavor, which is the next developmental step following the results in this article regarding hadron color.)

So, for what has heretofore been $\tau_i$, we now use the $3 \times 3$ SU(3) Gell-Mann generators via $\tau_i = \frac{1}{2}\lambda_i$ with $i = 1...8$, normalized to $\text{tr}(\tau_i)^2 = \frac{1}{2}$. We use $\text{diag}(\tau_8) = \frac{1}{2\sqrt{3}}(2, -1, -1)$, so that the $i = 1, 2, 3$ SU(2) subset is embedded in the lower-right portion of $\tau_i$. We may of course choose any labels we wish for these three exclusive states, so we may as well call these Red, Green and Blue, then see whether and how these can be made synonymous with the colored quark states of Quantum Chromodynamics (QCD). With $T$ denoting the transpose, we place these in the fundamental SU(3) representation with the explicit column vectors and consequent adjoints:

$$
\begin{aligned}
\psi_{(\sigma)} &\equiv \left| \tau_8 = +\tfrac{1}{2}\tfrac{2}{\sqrt{3}}; \tau_3 = 0 \right\rangle = \begin{pmatrix} \psi_R & 0 & 0 \end{pmatrix}^T; \quad \overline{\psi}_{(\sigma)} = \begin{pmatrix} \overline{\psi}_R & 0 & 0 \end{pmatrix} \\
\psi_{(\mu)} &\equiv \left| \tau_8 = -\tfrac{1}{2\sqrt{3}}; \tau_3 = +\tfrac{1}{2} \right\rangle = \begin{pmatrix} 0 & \psi_G & 0 \end{pmatrix}^T; \quad \overline{\psi}_{(\mu)} = \begin{pmatrix} 0 & \overline{\psi}_G & 0 \end{pmatrix} \\
\psi_{(\nu)} &\equiv \left| \tau_8 = -\tfrac{1}{2\sqrt{3}}; \tau_3 = -\tfrac{1}{2} \right\rangle = \begin{pmatrix} 0 & 0 & \psi_B \end{pmatrix}^T; \quad \overline{\psi}_{(\nu)} = \begin{pmatrix} 0 & 0 & \overline{\psi}_B \end{pmatrix}
\end{aligned}
\tag{29}
$$

Each of these $\psi$ is now a $3 \times 4$ column vector (ket) and each $\overline{\psi}$ a $3 \times 4$ row vector (bra), with the 3 owing to the YM SU(3) internal symmetry and the 4 owing to the four-components of Dirac wavefunctions and spinors. From here we will use (29) in (27), then in (28).

It is profoundly important that the Exclusion Principle taken together with the aforementioned "three-ness" of magnetic monopoles, combine to put exactly three fermions into the signal monopole for SU(N), and so lead directly to SU(N = 3) as the symmetry group required to establish three exclusive fermion states. Normally, SU(3) with R, G, B states is the starting point upon which QCD is founded. Here, in contrast, we can be entirely agnostic a priori about the N in SU(N), until we find that the inherent structure of a Yang–Mills signal magnetic monopole *requires* that fermions inside the monopole be placed into the fundamental representation of SU(3). This raises the prospect that QCD has its dynamic physical origins in the non-vanishing magnetic monopoles of Yang–Mills gauge theory. But again, for now, R, G and B are just labels: SU(3)$_{QCD}$ is an exact symmetry because gluons in its adjoint representation are massless. However, for example, early theories of baryon flavor similarly placed ($u$, $d$, $s$) into the fundamental representation of SU(3) with an approximate flavor symmetry which is distinct from the exact color symmetry of SU(3)$_{QCD}$. So, we must establish that this SU(3) group arising from the monopoles is truly synonymous with the exact SU(3) group of QCD, and not some independent SU(3).

## 11. The Yang–Mills Signal Magnetic Monopole Prior to Symmetry Breaking

Proceeding, we observe that at the center of each numerator in (27) are terms in which a wavefunction is immediately to the left of a differently labelled adjoint e.g., $\psi_{(\mu)}\overline{\psi}_{(\nu)}$, while in (28) we see the same-labelled e.g., $\psi_{(\sigma)}\overline{\psi}_{(\sigma)}$. In U(1) gauge theory these are $4 \times 4$ Dirac matrices, and in view of (29) these are $3 \times 3$ Yang–Mills matrices of $4 \times 4$ Dirac matrices. Specifically, focusing on the $\psi_{(\mu)}\overline{\psi}_{(\nu)}$, etc. "backbone" centering each numerator in (27), and carrying through antisymmetric signage between any pair of $\sigma, \mu, \nu$ indexes, we first observe using (29) that:

$$\psi_{(\mu)}\overline{\psi}_{(\nu)} - \psi_{(\nu)}\overline{\psi}_{(\mu)} + \psi_{(\nu)}\overline{\psi}_{(\sigma)} - \psi_{(\sigma)}\overline{\psi}_{(\nu)} + \psi_{(\sigma)}\overline{\psi}_{(\mu)} - \psi_{(\mu)}\overline{\psi}_{(\sigma)} = \begin{pmatrix} 0 & \psi_R\overline{\psi}_G & -\psi_R\overline{\psi}_B \\ -\psi_G\overline{\psi}_R & 0 & \psi_G\overline{\psi}_B \\ \psi_B\overline{\psi}_R & -\psi_B\overline{\psi}_G & 0 \end{pmatrix} \quad (30)$$

Then, we use (27) to flesh out the above, obtaining:

$$2ic\varepsilon_0 \mathrm{tr}\, p^{\sigma\mu\nu}/g^3 =$$
$$\begin{pmatrix} 0 & \dfrac{\partial^\nu\left(\overline{\psi}_{(\sigma)}\tau_i\gamma^\sigma\psi_R\overline{\psi}_G\tau_i\gamma^\mu\psi_{(\mu)}\right)}{\left(k_{\sigma,\mu}k^{\sigma,\mu}-m_{(\sigma,\mu)}{}^2c^2+i\varepsilon_{(\sigma,\mu)}\right)^2} & \dfrac{-\partial^\mu\left(\overline{\psi}_{(\sigma)}\tau_i\gamma^\sigma\psi_R\overline{\psi}_B\tau_i\gamma^\nu\psi_{(\nu)}\right)}{\left(k_{\{\nu,\sigma\}}k^{\{\nu,\sigma\}}-m_{(\nu,\sigma)}{}^2c^2+i\varepsilon_{(\nu,\sigma)}\right)^2} \\[2.5em] \dfrac{-\partial^\nu\left(\overline{\psi}_{(\mu)}\tau_i\gamma^\mu\psi_G\overline{\psi}_R\tau_i\gamma^\sigma\psi_{(\sigma)}\right)}{\left(k_{\sigma,\mu}k^{\sigma,\mu}-m_{(\sigma,\mu)}{}^2c^2+i\varepsilon_{(\sigma,\mu)}\right)^2} & 0 & \dfrac{\partial^\sigma\left(\overline{\psi}_{(\mu)}\tau_i\gamma^\mu\psi_G\overline{\psi}_B\tau_i\gamma^\nu\psi_{(\nu)}\right)}{\left(k_{\{\mu,\nu\}}k^\mu-m_{(\mu,\nu)}{}^2c^2+i\varepsilon_{(\mu,\nu)}\right)^2} \\[2.5em] \dfrac{\partial^\mu\left(\overline{\psi}_{(\nu)}\tau_i\gamma^\nu\psi_B\overline{\psi}_R\tau_i\gamma^\sigma\psi_{(\sigma)}\right)}{\left(k_{\{\nu,\sigma\}}k^{\{\nu,\sigma\}}-m_{(\nu,\sigma)}{}^2c^2+i\varepsilon_{(\nu,\sigma)}\right)^2} & \dfrac{-\partial^\sigma\left(\overline{\psi}_{(\nu)}\tau_i\gamma^\nu\psi_B\overline{\psi}_G\tau_i\gamma^\mu\psi_{(\mu)}\right)}{\left(k_{\{\mu,\nu\}}k^\mu-m_{\{\mu,\nu\}}{}^2c^2+i\varepsilon_{\{\mu,\nu\}}\right)^2} & 0 \end{pmatrix} \quad (31)$$

where $\left(k_\mu k^\mu - m_{(\mu)}{}^2c^2 + i\varepsilon_{(\mu)}\right)\left(k_\nu k^\nu - m_{(\nu)}{}^2c^2 + i\varepsilon_{(\nu)}\right) \equiv \left(k_{\{\mu,\nu\}}k^\mu - m_{(\mu,\nu)}{}^2c^2 + i\varepsilon_{(\mu,\nu)}\right)^2$, etc. is defined as a shorthand for the denominators simply to save space. Taking a second trace of the $3 \times 3$ matrix in (31), we see clearly that tr tr $p^{\sigma\mu\nu} = 0$.

If we look at any of the six off-diagonal entries in (31), for example, the term with a $\partial^\sigma\left(\overline{\psi}_{(\mu)}\tau_i\gamma^\mu\psi_G\overline{\psi}_B\tau_i\gamma^\nu\psi_{(\nu)}\right)$ numerator and a $\left(k_\mu k^\mu - m_{(\mu)}{}^2c^2 + i\varepsilon_{(\mu)}\right)\left(k_\nu k^\nu - m_{(\nu)}{}^2c^2 + i\varepsilon_{(\nu)}\right)$ denominator (second row, third column) while referring to (29), we see something of a mismatch, with two colors of fermion in the numerator and two massive vector boson propagators in the denominator. Fermions, of course, contain four degrees of freedom (spin up and down, particle and antiparticle), while massive vector bosons contain three degrees of freedom (two transverse, one longitudinal). So with both $\overline{\psi}_{(\mu)}$ and $\psi_{(\nu)}$, in this numerator there are two colors of fermion

totaling eight degrees of freedom, while with both $k_\mu k^\mu - m_{(\mu)}{}^2 c^2 + i\varepsilon_{(\mu)}$ and $k_\nu k^\nu - m_{(\nu)}{}^2 c^2 + i\varepsilon_{(\nu)}$ in the denominator there are two massive vector boson propagator denominators representing a total of six degrees of freedom.

Moreover, SU(3)$_{QCD}$ requires massless gluons, whereas the denominators in (31) are all for massive vector bosons. So this looks like an approximate SU(3) flavor-type rather than an exact SU(3) color symmetry. Moreover, (31) has vanishing trace. Furthermore, knowing that all particle propagators have the general form $i\Sigma_{\text{spins}} / (p^2 - m^2 + i\varepsilon)$ with spin sum $\Sigma_{\text{spins}}$ being the completeness relation, we see that $\psi_G \overline{\psi}_B$ at the center of this numerator looks like a spin sum $\Sigma_s u\overline{u} = p + m$, but is not. This is because $\psi_G$ and $\psi_B$ in $\psi_G \overline{\psi}_B$ are not the same fermion but are two differently-colored fermions. This final point indicates the way forward, because if we can turn $\psi_G$ and $\psi_B$ in $\psi_G \overline{\psi}_B$ into the same fermion, we can use this as a fermion spin sum. Then, having a spin sum in the numerator and two massive boson propagators in the denominator, we can shuttle a degree of freedom from a boson to a fermion to simultaneously produce a fermion propagator and a massless boson propagator. This entails a form of spontaneous symmetry breaking which starts with (27) then breaks symmetry using (28), because the backbone of (28) does have the requisite same-fermion terms $\psi_{(\sigma)} \overline{\psi}_{(\sigma)}$, $\psi_{(\mu)} \overline{\psi}_{(\mu)}$ and $\psi_{(\nu)} \overline{\psi}_{(\nu)}$ needed to use the completeness relation.

## 12. Spontaneously Breaking Symmetry inside the Yang–Mills Signal Magnetic Monopole

Keeping the fermion state definitions (29) exactly as is, we now examine the "backbone" of (28) formed by the three terms $\psi_{(\sigma)} \overline{\psi}_{(\sigma)}$, $\psi_{(\mu)} \overline{\psi}_{(\mu)}$ and $\psi_{(\nu)} \overline{\psi}_{(\nu)}$. Each wavefunction $\psi(x^\mu, p^\mu) = u(p^\mu) \exp(-ip_\sigma x^\sigma)$ has adjoint $\overline{\psi} = \overline{u} \exp(ip_\sigma x^\sigma)$, and because these are back-to-back with no $\gamma^\alpha$ between them, their $p_\sigma$ are the same. Here, however, unlike with (30), because these are same-labelled with the same $p_\sigma$, we may not only write $\psi\overline{\psi} = u\overline{u}$, but given that $\Sigma_s u\overline{u} = p + m$ for the spin sum over fermion particle states, we may use $u\overline{u}$ as the basis for a spin sum which can lead to a fermion propagator. So, if we additionally take the sum $\Sigma_s$ over particle spins, what we find in contrast to (30) is the now-diagonalized backbone:

$$\Sigma_s\left(\psi_{(\sigma)} \overline{\psi}_{(\sigma)} + \psi_{(\mu)} \overline{\psi}_{(\mu)} + \psi_{(\nu)} \overline{\psi}_{(\nu)}\right)$$
$$= \Sigma_s \begin{pmatrix} \psi_R \overline{\psi}_R & 0 & 0 \\ 0 & \psi_G \overline{\psi}_G & 0 \\ 0 & 0 & \psi_B \overline{\psi}_B \end{pmatrix} = \Sigma_s \begin{pmatrix} u_R \overline{u}_R & 0 & 0 \\ 0 & u_G \overline{u}_G & 0 \\ 0 & 0 & u_B \overline{u}_B \end{pmatrix} = \begin{pmatrix} p_R + m_R & 0 & 0 \\ 0 & p_G + m_G & 0 \\ 0 & 0 & p_B + m_B \end{pmatrix} \tag{32}$$

Next, we use (32) to flesh out (28), as we did with (30) for (27) to obtain (31). In doing so, we take the spin sums over all of the fermions inside of (28), which implies taking $\Sigma_s p^{\sigma\mu\nu}$ over the entire signal monopole system as well. Consequently, with this spin sum included and $c = 1$:

$$2ic\varepsilon_0 \text{tr}\, \Sigma_s p^{\sigma\mu\nu} / g^3 =$$
$$\begin{pmatrix} \partial^\sigma \dfrac{\overline{\psi}_{(\sigma)} \tau_i \gamma^{[\mu}(p_R + m_R)\tau_i \gamma^{\nu]} \psi_{(\sigma)}}{\left(k_\sigma k^\sigma - m_{(\sigma)}{}^2 + i\varepsilon_{(\sigma)}\right)^2} & 0 & 0 \\ 0 & \partial^\mu \dfrac{\overline{\psi}_{(\mu)} \tau_i \gamma^{[\nu}(p_G + m_G)\tau_i \gamma^{\sigma]} \psi_{(\mu)}}{\left(k_\mu k^\mu - m_{(\mu)}{}^2 + i\varepsilon_{(\mu)}\right)^2} & 0 \\ 0 & 0 & \partial^\nu \dfrac{\overline{\psi}_{(\nu)} \tau_i \gamma^{[\sigma}(p_B + m_B)\tau_i \gamma^{\mu]} \psi_{(\nu)}}{\left(k_\nu k^\nu - m_{(\nu)}{}^2 + i\varepsilon_{(\nu)}\right)^2} \end{pmatrix} \tag{33}$$

This is the diagonalized counterpart of (31). Because of this, we may take another trace which, in contrast to $\text{tr}\,\text{tr}(p^{\sigma\mu\nu}) = 0$ in (31), is non-vanishing, namely:

$$2ic\varepsilon_0 \text{tr}\,\text{tr}\, \Sigma_s p^{\sigma\mu\nu} / g^3 =$$
$$\left(\partial^\sigma \dfrac{\overline{\psi}_{(\sigma)} \tau_i \gamma^{[\mu}(p_R + m_R)\tau_i \gamma^{\nu]} \psi_{(\sigma)}}{\left(k_\sigma k^\sigma - m_{(\sigma)}{}^2 + i\varepsilon_{(\sigma)}\right)^2} + \partial^\mu \dfrac{\overline{\psi}_{(\mu)} \tau_i \gamma^{[\nu}(p_G + m_G)\tau_i \gamma^{\sigma]} \psi_{(\mu)}}{\left(k_\mu k^\mu - m_{(\mu)}{}^2 + i\varepsilon_{(\mu)}\right)^2} + \partial^\nu \dfrac{\overline{\psi}_{(\nu)} \tau_i \gamma^{[\sigma}(p_B + m_B)\tau_i \gamma^{\mu]} \psi_{(\nu)}}{\left(k_\nu k^\nu - m_{(\nu)}{}^2 + i\varepsilon_{(\nu)}\right)^2}\right) \tag{34}$$

Next, we focus on the Yang–Mills aspect of the numerators irrespective of Dirac matrices, which we can do because $\gamma^\mu$ and $\tau_i$ with $[\gamma^\mu, \tau_i] = 0$ are independent operators acting on independent parts of each Nx4 bra or ket in (29). We find the three structural combinations $\overline{\psi}_{(\sigma)}\tau_i\tau_i\psi_{(\sigma)}$, $\overline{\psi}_{(\mu)}\tau_i\tau_i\psi_{(\mu)}$ and $\overline{\psi}_{(\nu)}\tau_i\tau_i\psi_{(\nu)}$. So we use $\tau_i = \frac{1}{2}\lambda_i$ to calculate eight column vectors $\tau_i\psi_{(\sigma)}$, eight $\tau_i\psi_{(\mu)}$, and eight $\tau_i\psi_{(\nu)}$. The adjoint $\overline{\psi}_{(\sigma)}\tau_i$, $\overline{\psi}_{(\mu)}\tau_i$ and $\overline{\psi}_{(\nu)}\tau_i$ are the Hermitian conjugates of these. The net result, easily confirmed, is: (For SU(N) in general, the factor $f = 4/3$ shown for SU(3) in (35) below, is equal to $\frac{1}{2}$ times the number of states in the adjoint representation over the number of states in the fundamental representation, that is $f = \frac{1}{2}(N^2 - 1)/N$. This is also equal to the magnitude of the principal Casimir operator for any SU(N), that is, $f\,\mathrm{Id} = \tau^2 = \Sigma_i \tau_i^2$.)

$$\overline{\psi}_{(\sigma)}\tau_i\tau_i\psi_{(\sigma)} = \frac{4}{3}\overline{\psi}_R\psi_R; \quad \overline{\psi}_{(\mu)}\tau_i\tau_i\psi_{(\mu)} = \frac{4}{3}\overline{\psi}_G\psi_G; \quad \overline{\psi}_{(\nu)}\tau_i\tau_i\psi_{(\nu)} = \frac{4}{3}\overline{\psi}_B\psi_B \tag{35}$$

We then use (35) in (34) and multiply through by $i$ to obtain:

$$-c\varepsilon_0 \mathrm{tr}\,\mathrm{tr}\,\Sigma_s p^{\sigma\mu\nu} =$$
$$i\frac{2}{3}g^3\left(\partial^\sigma \frac{\overline{\psi}_R\gamma^{[\mu}(p_R + m_R)\gamma^{\nu]}\psi_R}{\left(k_\sigma k^\sigma - m_{(\sigma)}^2 + i\varepsilon_{(\sigma)}\right)^2} + \partial^\mu \frac{\overline{\psi}_G\gamma^{[\nu}(p_G + m_G)\gamma^{\sigma]}\psi_G}{\left(k_\mu k^\mu - m_{(\mu)}^2 + i\varepsilon_{(\mu)}\right)^2} + \partial^\nu \frac{\overline{\psi}_B\gamma^{[\sigma}(p_B + m_B)\gamma^{\mu]}\psi_B}{\left(k_\nu k^\nu - m_{(\nu)}^2 + i\varepsilon_{(\nu)}\right)^2}\right) \tag{36}$$

Above, the mismatch between numerators containing fermions and denominators with massive vector boson propagators becomes crystalized. However, so too does the solution: In the numerators we have spin sums $p + m$ for fermions, mismatched in the denominators with a pair $\left(k^2 - m^2 + i\varepsilon\right)\left(k^2 - m^2 + i\varepsilon\right)$ of massive vector boson propagators. Indeed, the reason we carefully respectively established the energy-momentum notation conventions $p^\mu$, $k^\mu$ and $q^\mu$ for fermions, massive vector bosons and massless vector bosons at the start of Section 6, was so when we arrived at (36) this mismatch would be clear.

Now, with the $i$, let us work with the (36) term operated upon. e.g., by $\partial^\sigma$, ditto for the others. In the denominator we revert to indexes/labels showing the currents $j^\mu$ and $j^\nu$ which were initially associated with these fermions in (26), which appear in the denominator of (27). This $\left(k_\mu k^\mu - m_{(\mu)}^2 + i\varepsilon_{(\mu)}\right)\left(k_\nu k^\nu - m_{(\nu)}^2 + i\varepsilon_{(\nu)}\right)$ represents a total of six degrees of freedom, three for each of two massive vector bosons. Although the massive bosons were introduced by hand at (15), it is well-known how to give a renormalizable mass to these via $\mathcal{L} = \partial^\sigma\phi * \partial_\sigma\phi + g^2 G^\sigma G_\sigma \phi * \phi$ using the Higgs mechanism. We then break symmetry by releasing one degree of freedom from the $k_\mu k^\mu - m_{(\mu)}^2 + i\varepsilon_{(\mu)}$ denominator and shuttling this over to be swallowed by the $k_\nu k^\nu - m_{(\nu)}^2 + i\varepsilon_{(\nu)}$ denominator, or vice versa—it does not matter. Doing so, we demote $k^\mu \mapsto q^\mu$ into a *massless* vector boson such as a QCD gluon while consequently setting $m_{(\mu)} = 0$. Simultaneously, we promote $k^\nu \mapsto p^\nu$ to the energy momentum of the fermion and $m_{(\nu)} \mapsto m_R$ and $\varepsilon_{(\nu)} \mapsto \varepsilon_R$ to the mass and $\varepsilon$ of the R fermion. Finally, we rename denominator indexes back to those for $\partial^\sigma$ in (36). We summarize this symmetry breaking as follows:

$$i\frac{\overline{\psi}_R\gamma^{[\mu}(p_R + m_R)\gamma^{\nu]}\psi_R}{\left(k_\mu k^\mu - m_{(\mu)}^2 + i\varepsilon_{(\mu)}\right)\left(k_\nu k^\nu - m_{(\nu)}^2 + i\varepsilon_{(\nu)}\right)} \xrightarrow{\text{break symmetry}} \frac{1}{q_\sigma q^\sigma + i\varepsilon_{(\sigma)}} \frac{\overline{\psi}_R\gamma^{[\mu}i(p_R + m_R)\gamma^{\nu]}\psi_R}{p_{R\sigma}p_R^\sigma - m_R^2 + i\varepsilon_R} \tag{37}$$

Above, the original 3 + 3 = 6 degrees of freedom in the denominator are maintained, but redistributed to 2 + 4 = 6 degrees of freedom, two for the now-massless vector boson labelled with $\sigma$, and four for the red fermion. Doing the same and accounting for all three additive terms in (36) which now contain the three R, G, B fermion states, we multiply by 3 so that 9 + 9 = 18 degrees of freedom at the outset in six massive vector boson propagators have become redistributed as 6 + 12 = 18 into three massless vector boson and three massive fermion propagators. If we suppose that the massive vector bosons on the left of (37) originally come from a scalar field via the Higgs mechanism, then (37) is a later step in a "cascade" wherein a degree of freedom is first passed from a

scalar to a massless vector boson to make the latter massive, then is passed from the vector boson to a fermion whereby the vector boson reverts to being massless. Mindful that $(p + m)(p - m) = p_\sigma p^\sigma - m^2$, the upshot is that we now have a fermion propagator with the $+i\varepsilon$ prescription sitting in the middle of (37).

Finally, replicating (37) with necessary reindexing and relabeling twice over then inserting into (36) we obtain:

$$-c\varepsilon_0 \text{tr tr } \Sigma_s p^{\sigma\mu\nu} = \frac{2}{3}g^3 \left( \begin{array}{l} \frac{1}{q_\sigma q^\sigma + i\varepsilon_{(\sigma)}} \partial^\sigma \frac{\overline{\psi}_R \gamma^{[\mu} i(p_R + m_R)\gamma^{\nu]}\psi_R}{p_{R\sigma}p_R{}^\sigma - m_R{}^2 + i\varepsilon_R} \\ + \frac{1}{q_\mu q^\mu + i\varepsilon_{(\mu)}} \partial^\mu \frac{\overline{\psi}_G \gamma^{[\nu} i(p_G + m_G)\gamma^{\sigma]}\psi_G}{p_{G\mu}p_G{}^\mu - m_G{}^2 + i\varepsilon_G} \\ + \frac{1}{q_\nu q^\nu + i\varepsilon_{(\nu)}} \partial^\nu \frac{\overline{\psi}_B \gamma^{[\sigma} i(p_B + m_B)\gamma^{\mu]}\psi_B}{p_{B\nu}p_B{}^\nu - m_B{}^2 + i\varepsilon_B} \end{array} \right) \quad (38)$$

This is now the Yang–Mills magnetic monopole with symmetry broken in two stages: First, by taking (27) where each fermion is inserted into the monopole via a current $j^\sigma$, $j^\mu$ or $j^\nu$ and turning it into (28) whereby the fermions now form part of an *indivisible* monopole system, no longer distinguishing fermions based on the index of their initial current. In this stage, (31) is diagonalized and turned into (33). Second, by using (37) to transfer longitudinal vector boson degrees of freedom over to fermions thus rendering the remaining vector bosons massless and revealing complete fermion propagators. Now, with the vector bosons being massless, the SU(3) symmetry introduced at (29) becomes an exact symmetry like that of SU(3)$_{\text{QCD}}$, rather than the approximate symmetry of SU(3) flavor, bringing us closer to these R, G, B being true QCD states.

## 13. Incorporating the Massless Vector Boson Propagators into the Fermion Normalizations

We can further simplify (38) by suitable normalization of the three fermion spinors. Often, a covariant normalization with an energy-dimensioned $N^2 = E + mc^2$ is employed for this. If we write this with $c = 1$ as $1 = N_{R,G,B}{}^2 / (E + m)_{R,G,B}$ and place this with the corresponding label in front of each of the three terms in (38), we end up with $N_R{}^2 / (E + m)_R \left( q_\alpha q^\alpha + i\varepsilon_{(\alpha)} \right)$ in the top line, ditto for the others. Then, because the YM monopole is an indivisible system, we may now choose $N'_R{}^2 / (E + m)_R \left( q_\alpha q^\alpha + i\varepsilon_{(\alpha)} \right) \equiv 1$ as a modified normalization in which $N'_R{}^2$ scales with the $q_\alpha q^\alpha + i\varepsilon_{(\alpha)}$ of a massless vector boson within the overall system. Doing this, (38) simplifies to:

$$-c\varepsilon_0 \text{tr tr } \Sigma_s p^{\sigma\mu\nu} =$$
$$\frac{2}{3}g^3 \left( \partial^\sigma \frac{\overline{\psi}_R \gamma^{[\mu} i(p_R + m_R)\gamma^{\nu]}\psi_R}{p_{R\sigma}p_R{}^\sigma - m_R{}^2 + i\varepsilon_R} + \partial^\mu \frac{\overline{\psi}_G \gamma^{[\nu} i(p_G + m_G)\gamma^{\sigma]}\psi_G}{p_{G\mu}p_G{}^\mu - m_G{}^2 + i\varepsilon_G} + \partial^\nu \frac{\overline{\psi}_B \gamma^{[\sigma} i(p_B + m_B)\gamma^{\mu]}\psi_B}{p_{B\nu}p_B{}^\nu - m_B{}^2 + i\varepsilon_B} \right) \quad (39)$$

This becomes our final expression for the signal (non-perturbative) Yang–Mills magnetic monopole, which clearly embeds a propagator for each of its three fermions. Now we are ready to show why these monopoles have the same color and confinement properties as baryons, with interactions mediated by entities which have the same color properties as mesons.

## 14. Yang–Mills Magnetic Monopoles Have the Color-Neutral Singlet Wavefunction of Baryons, and Interact via Objects with the Color-Neutral Singlet Wavefunction of Mesons

We know that the RGB SU(3) symmetry in the monopole (39) is exact, because the vector bosons were made massless at (37). We know that in its unperturbed signal state, this monopole contains exactly three fermions. We know too, that this monopole $p^{\sigma\mu\nu}$ is a third rank antisymmetric tensor. However, a key step in going from (27) to (28) in the first symmetry breaking stage, was to assign each fermion, not to the current which caried it into the monopole, but to the index of the partial derivative commonly operating on those fermions once they were already inside the monopole. This led to the associations $\sigma \sim R$, $\mu \sim G$ and $\nu \sim B$, as very clearly seen in (39).

So, writing out the antisymmetry of the monopole indexes and relating these to their color associations once the fermions are all inside the monopole, the wavefunction symmetry of the now-indivisible monopole system may be schematically represented by:

$$p^{\sigma\mu\nu} \sim \sigma\mu\nu - \sigma\nu\mu + \mu\nu\sigma - \mu\sigma\nu + \nu\sigma\mu - \nu\mu\sigma \sim RGB - RBG + GBR - GRB + BRG - BGR \tag{40}$$

This is precisely the antisymmetric color-neutral singlet wavefunction of a baryon, see eq. [2.70] of [16]. Indeed, one can argue that the antisymmetric indexes in $p^{\sigma\mu\nu}$ should have been a tip-off that YM magnetic monopoles would make good baryons. Though individual fermions and vector bosons inside the monopole carry color charges, the entire monopole system is a color singlet.

Next, we return to the differential forms relation $c\mu_0 \iiint p = -igG^2 \neq 0$ of (14b) and again ask as we did at the end of Section 4: what are these $G^2 = \frac{1}{2}[G_\mu, G_\nu] dx^\mu dx^\mu$ entities which *do* net flow across a YM magnetic monopole surface? Lowering all free indexes in (39), then using $p = \frac{1}{3!} p_{\sigma\mu\nu} dx^\sigma dx^\mu dx^\nu$ to obtain the monopole 3-form, and using the antisymmetry among each infinitesimal element in $dx^\sigma dx^\mu dx^\nu$ along with index renaming as needed, and also using $c^2\varepsilon_0\mu_0 = 1$ while reconnecting this to (13b), we obtain:

$$-c\varepsilon_0 \text{tr tr} \Sigma_s p = ic\varepsilon_0 \frac{1}{3!} \text{tr tr} \Sigma_s p_{\sigma\mu\nu} dx^\sigma dx^\mu dx^\nu = c^2\varepsilon_0^2 d \text{ tr tr} \Sigma_s g G^2$$
$$\partial_\sigma \left\{ \frac{2}{9}\frac{1}{2}g^3 \left( \frac{\overline{\psi}_R \gamma_{[\mu} i(p_R + m_R)\gamma_{\nu]}\psi_R}{p_{R\sigma}p_R{}^\sigma - m_R{}^2 + i\varepsilon_R} + \frac{\overline{\psi}_G \gamma_{[\mu} i(p_G + m_G)\gamma_{\nu]}\psi_G}{p_{G\mu}p_G{}^\mu - m_G{}^2 + i\varepsilon_G} + \frac{\overline{\psi}_B \gamma_{[\mu} i(p_B + m_B)\gamma_{\nu]}\psi_B}{p_{B\nu}p_B{}^\nu - m_B{}^2 + i\varepsilon_B} \right) dx^\mu dx^\nu \right\} dx^\sigma \tag{41}$$

We then take the triple integral of all sides and apply (6) via $\int_M dp = \int_{\partial M} p$, to find:

$$-c\varepsilon_0 \iiint \text{tr tr} \Sigma_s\, p = ic^2\varepsilon_0^2 \iiint d\{\text{tr tr} \Sigma_s g G^2\} = ic^2\varepsilon_0^2 \text{tr tr} \Sigma_s\, gG^2$$
$$= \frac{2}{9}g^3\frac{1}{2}\left( \frac{\overline{\psi}_R \gamma_{[\mu} i(p_R + m_R)\gamma_{\nu]}\psi_R}{p_{R\sigma}p_R{}^\sigma - m_R{}^2 + i\varepsilon_R} + \frac{\overline{\psi}_G \gamma_{[\mu} i(p_G + m_G)\gamma_{\nu]}\psi_G}{p_{G\mu}p_G{}^\mu - m_G{}^2 + i\varepsilon_G} + \frac{\overline{\psi}_B \gamma_{[\mu} i(p_B + m_B)\gamma_{\nu]}\psi_B}{p_{B\nu}p_B{}^\nu - m_B{}^2 + i\varepsilon_B} \right) dx^\mu dx^\nu \tag{42}$$

Above, the Gaussian integration has removed the $\partial_\sigma$ operator from (41). Extracting the integrands from the surface integrals in (42) and using (9) written as $-(F - dG) = igG^2$, then using $G^2 = \frac{1}{2}[G_\mu, G_\nu] dx^\mu dx^\mu$, restoring all spacetime indexes and removing $dx^\mu dx^\nu$, we find:

$$-c^2\varepsilon_0^2 \text{tr tr} \Sigma_s \left( F_{\mu\nu} - \partial_\mu G_\nu + \partial_\nu G_\mu \right) = ic^2\varepsilon_0^2 \text{ tr tr} \Sigma_s g[G_\mu, G_\nu]$$
$$= \frac{2}{9}g^3\left( \frac{\overline{\psi}_R \gamma_{[\mu} i(p_R + m_R)\gamma_{\nu]}\psi_R}{p_{R\sigma}p_R{}^\sigma - m_R{}^2 + i\varepsilon_R} + \frac{\overline{\psi}_G \gamma_{[\mu} i(p_G + m_G)\gamma_{\nu]}\psi_G}{p_{G\mu}p_G{}^\mu - m_G{}^2 + i\varepsilon_G} + \frac{\overline{\psi}_B \gamma_{[\mu} i(p_B + m_B)\gamma_{\nu]}\psi_B}{p_{B\nu}p_B{}^\nu - m_B{}^2 + i\varepsilon_B} \right) \tag{43}$$

By inspection, (42) and (43) have the respective schematic color wavefunctions:

$$\iiint p \sim \oiint G^2 \sim \left( \oiint \overline{R}R + \overline{G}G + \overline{B}B \right) \neq 0; \ \ G^2 = \frac{1}{2}[G_\mu, G_\nu]dx^\mu dx^\mu \sim \overline{R}R + \overline{G}G + \overline{B}B \tag{44}$$

This is precisely the required symmetric color-neutral singlet wavefunction for a meson. So in contrast to the U(1) magnetic monopoles of Maxwell for which there is no net magnetic field flux across any closed surface, there is a net flux of "chromo-magnetic" fields across the surface surrounding a Yang–Mills magnetic monopole, namely the $G^2$ first identified at (14b). However, as seen in (44), these $G^2$ objects are color-neutral, so there is still no flow of net color charge across any closed surface surrounding or within the monopole. We now see these have the required color-neutral wavefunction of mesons known to mediate baryon interactions.

Accordingly, we conclude that Yang–Mills magnetic monopoles have the antisymmetric color-neutral singlet wavefunction of baryons, and objects which net flow across their closed surfaces have the symmetric color-neutral singlet wavefunction of mesons. Together, these are the hadrons.

## 15. Act of Confinement: Dynamical Hadronization from Maxwell's Yang–Mills Equations

In his review of the MIT bag model in Section 18 of [6], Close reviews Gauss's theorem for electric charge—contained in $c\mu_0 \iiint *j = \oiint *F$ from (7a)—then "consider[s] the chromodynamics case which is analogous to" Gauss's theorem. He states: "if the demand that no quark current crosses the boundary is supplemented by the demand that colour gluons are also confined then Gauss's theorem implies that the system have zero colour charge." He continues that in the bag model, "the introduction of a pressure $B$ that counterbalances the flow of colour flux automatically requires the system to be colour neutral. If colour symmetry is exact then the system must be a colour singlet." This is precisely true of (39): its color symmetry is exact because the symmetry breaking in (37) made its gauge bosons massless, and it is a color singlet.

Close then makes the critical points, emphasis added, that "*quark confinement arises as a result of colour confinement*," and that the bag model "imposition ad hoc of a boundary condition that confines the coloured gluons has, by Gauss, confined the coloured quarks." Importantly, he concludes that "*a dynamical origin for this boundary condition has not been presented*" by the bag model. Or it appears, by any other theory to date.

Here, (43) and (44) demonstrate "by Gauss" that the objects which net flow across the monopole surface are color-neutral. This means, conversely, that objects which are not color neutral, i.e., which do have a color charge, do *not* net flow across the surface but are confined. (Because the confined objects are those with non-neutral net color, there is nothing in this result which prevents the electroweak photon, $W^\pm$ and $Z$ from flowing across the monopole surface, because these are color-neutral.) Inside these signal monopoles, each of the three fermions has a net color charge in the fundamental representation of an SU(3) gauge group which is exact because its vector bosons are massless, and each of these massless vector bosons has a net bi-colored charge in the adjoint representation of SU(3). We therefore conclude that these fermions and massless gauge bosons are confined. Consequently, we further conclude that: the fermions in (39) are quarks; the now-massless gauge bosons are gluons; the signal monopole (39) is a baryon in a non-perturbative state with all "noise" filtered out; and the $G^2$ object (43) which net flows across the monopole-now-baryon surface is a meson.

Crucially, this is not an ad hoc result. It has a "*dynamical origin*" in the very fundamental physics of Maxwell's equations extended to the non-commuting gauge fields of Yang and Mills, coupled with Dirac's quantum theory of fermions [12] and the requirement that each fermion in a system such as an atom or a nucleus or a nucleon or a baryon must occupy an exclusive quantum state. There is nothing new or unsettled in any of the individual elements which are combined to reach this dynamical result. What is new is simply understanding how these all combine together to produce the hadronic phenomenology of QCD, and how the rank-3 antisymmetric structure of a magnetic monopole is dynamically responsible for SU(3)—not SU(2) or SU(4) or anything else—being the gauge group underlying hadronic physics. So, we do not need to postulate SU(3)$_{\text{QCD}}$ as has been done ever since Gell-Mann [17] and Zweig [18] first discovered the quark model. The Yang–Mills magnetic monopoles dynamically make that postulate for us, all by themselves.

In cosmology, it is widely believed that hadronization occurred shortly after the Big Bang when the quark-gluon plasma cooled to the temperature below which free quarks and gluons cannot exist. In view of all the above, we can now identify the symmetry breaking of Section 12 with hadronization of a free quark and gluon plasma believed to exist only at ultra-high GUT energies above ~$10^{15}$ GeV, not far below the Planck scale $E_P = \sqrt{hc^5/G} \approx 1.220 \times 10^{19}$ GeV. Specifically, the signal monopole obtained in (31) prior to symmetry breaking has tr tr $p^{\sigma\mu\nu} = 0$ and is associated with energies above $10^{15}$ GeV where quarks are free and can mingle with leptons, and where baryons with confined quarks and gluons are not yet formed. The signal monopole obtained in (39) after symmetry breaking has tr tr $\Sigma_s p^{\sigma\mu\nu} \neq 0$ and is associated with lower energies where free quarks and gluons no longer exist but are confined in color-neutral hadrons. So, the symmetry breaking to go from (27) to (28) in concert with (37) is now seen to take place at some energy $E_X$ a few orders of magnitude below the Planck energy. The pre-symmetry breaking (27) in which quarks are labelled with the spacetime indexes of the current densities which carry

them into the monopole shows the pre-hadronization baryon above $E_X$, which at (31) has tr tr $p^{\sigma\mu\nu} = 0$. We shall refer to these as the "plasma" labels. The post-break (28) in which the monopole is made indivisible with quarks now labelled independently from their current of origin shows the baryon below $E_X$ once hadronization is complete, which at (39) now has tr tr $\Sigma_s p^{\sigma\mu\nu} \neq 0$ with confined color. We shall refer to these as the "confinement" labels. The symmetry breaking from (27) to (28), which via (37) takes us from (31) to (39), then becomes synonymous with $E_X$-scale hadronization which is dynamical, not ad hoc. This is what Close refers to as the "act of confinement."

### 16. What Is a Baryon, and Who Ordered That?

If we overlook Rabi's question, and take baryons and mesons as an empirical given without inquiring about their theoretical genesis, then we are pressed into asking questions about how quarks remain confined inside of baryons without knowing what baryons really are, and perhaps, to engineer ad hoc models of attractive and repulsive forces which yield quark and color confinement from a data fitting exercise, rather than obtaining the dynamic understanding referenced by Close [6]. We are also pressed with only incomplete knowledge to ask how the transition takes place from a quark/gluon plasma at ultra-high energies and temperatures, to baryons with confined, asymptotically free quarks observed at ordinary energies and temperatures. Nor do we have a basis for truly understanding the quark–antiquark pairs which form the mesons that mediate baryon interactions, and which permeate the space inside a baryon. Further, although we do know that QCD stems from an exact SU(3) color symmetry, we do not know *why* SU(3), as opposed to some other gauge group, provides this basis for strong interaction physics. All the while, lattice QCD provides only convincing numeric computation, not fully analytic proof, that our theories about strong interactions are correct, because of the perceived intractability of analytically studying the foregoing phenomena. This has produced a substantial body of literature exemplified by [19–25], all seeking to answer these questions about confinement and hadronization and quark–antiquark objects and even the core nature of strong interactions, without the benefit of knowing what baryons really are. However, if we do know the true theoretical origin of baryons, then hadronization leading to confinement and to baryon interactions via mesons and even to SU(3) QCD itself ought to flow effortlessly from the natural inherent properties of these baryons.

By attending to Rabi's question as has been done here, and discovering that baryons are in fact the magnetic monopoles of Yang–Mills gauge theory populated by the fermions in the electric charge sources of Yang–Mills gauge theory, Maxwell's equations—extended from U(1) abelian to SU(N) Yang–Mills gauge theories—are naturally placed at the center of how we describe the behaviors of these baryons. Then, the integral form of Maxwell's equations, particularly Gauss's law for magnetism, simply becomes a statement of what does and does not flow across any closed surface around or within a baryon, in the basic spirit of the MIT bag model.

Consequently, one can examine quark and color confinement by simply deciphering what these Maxwell/Yang–Mills equations tell us about net fluxes across monopole boundaries, without having to construct elaborate models of attractive and repulsive forces engineered to bring about confinement. Specifically, in U(1) electrodynamics, Gauss's law implies the inverse-square force of Coulomb's law. So by approaching the confinement of quarks via what does and does not flow across closed surfaces, and finding that there is no flow of net color but there is a flow of colorless antisymmetric mesons in accordance with (44), we are implicitly ensuring that the forces inside the baryon linked to these Gauss's law surfaces will operate to confine the quarks. That is, if we start with a theoretical monopole baryon in which no net color can ever cross a closed Gaussian surface around, partially through, or entirely within this baryon due to the surface flux properties inherent to monopoles which have been known since the time of Maxwell, then we are assured that the attractive and repulsive forces associated with this Yang–Mills extension of Gauss's law for magnetism will balance precisely as needed to confine net color without ad hoc engineering. So, because each quark does have a net color charge and each gluon a net bi-colored charge, this means that the quarks and gluons are naturally confined.

In sum, once we engage Rabi's question and understand that baryons are the Maxwell magnetic monopoles of Yang–Mills gauge theory, the question why quarks remain confined obtains a very simple and natural answer: they are confined because this is how Yang–Mills magnetic monopoles naturally behave. No more and no less. YM monopoles naturally bar net fluxes of color not only across what one might define as an infrared confinement boundary near about 1 fermi (the rough size of a baryon), but also across any smaller closed sub-surfaces wholly or partially inside of the baryon. Because quarks and gluons have net color and bi-color charges, with net color confined in this way, quarks and gluons will also be confined in the very same way.

Moreover, with such an understanding, rather than hypothesize a supersymmetry between bosons and fermions which has long been conjectured but never obtained any empirical support, we discover that hadronization employs a type of spontaneous symmetry breaking where longitudinal degrees of freedom in massive vector bosons are transferred over to fermions during a type of symmetry breaking not dissimilar to the Higgs mechanism which moves degrees of freedom from scalar bosons to vector bosons. This mechanism, particularly via the electroweak *W* and *Z* bosons [26–28] and later through the direct observation of the Higgs boson [29], has found clear experimental support and so gained wide acceptance. Such hadronization symmetry breaking makes those vector bosons massless and simultaneously renders the underlying symmetry of fermions in the fundamental representation of SU(3) exact, which is the foundation [17], [18] of settled QCD. This all proceeds in the ordinary four dimensions of spacetime without the need for additional space dimensions and compactification down to four spacetime dimensions, which originated with the work of Kaluza and Klein [30–32].

With this understanding, SU(3) QCD itself becomes no longer just a highly successful postulate both theoretically and experimentally; rather it comes to rest, fully intact, on a deeper dynamic foundation: Because the ground state "signal" monopole is a system containing precisely three fermions, the Exclusion Principle naturally places these fermions into the fundamental representation of SU(3), which becomes exact once symmetry is broken and hadronization of the plasma is complete. What ties this all together very simply, is the fact that magnetic monopoles are third-rank antisymmetric tensors as first elaborated by Einstein in his seminal paper [8]. So, when these monopoles are promoted from abelian to non-abelian status via Yang–Mills gauge theory [1], then populated with source currents from the inverted Maxwell's YM charge equation, they not only contain exactly three fermions in their ground state, but they become color-neutral, having the three-quark antisymmetric wavefunction (40) which carries a fingerprint identical to that of the generally covariant index structure of the magnetic monopole.

Indeed, the third-rank antisymmetric tensor structure of magnetic monopoles having no net magnetic charge and permitting no net magnetic field flux through closed spatial surfaces is perhaps the strongest clue that these monopoles are naturally suited to replicate the three-quark antisymmetric net-colorless wavefunction of a baryon with no net color flux through any closed surface. Thus, when understood through the integral formulation of Gauss's law for magnetism which again motivates the MIT bag model, confinement boils down to the Yang-Mills magnetic monopoles barring a net flow of color through closed surfaces in the same way that U(1) abelian monopoles bar a net flow of magnetic fields through such surfaces. Meanwhile, hadronic interactions occur via colorless mesons which are enabled to and indeed do net flow through closed surfaces around these monopoles, while inside of monopoles all fluxes through closed sub-surfaces must likewise be color-neutral. The general result is that net color may not flow through closed Gaussian surfaces anywhere, analogously to magnetic fields in U(1) electrodynamics, and any net fluxes which do occur must be by color-neutral objects such as mesons and jets.

It is also important to be mindful that the complete magnetic monopole which corresponds to the physically observed baryons with all non-linear perturbative behaviors included is that of the infinitely recursive (24) which contains both the monopole "signal plus noise," that (39) shows the "signal" rendition of this monopole with all "noise" removed, and that this "signal" monopole (39) is what directly reveals the color-neutral antisymmetric wavefunction (40) of a baryon, and following

Gaussian integration, the color-neutral symmetric wavefunction (44) of a meson for fluxes through closed surfaces. The is the analytical way to understand what is ordinary studied using the numerical approach of lattice QCD. Indeed, the perceived intractability of analytical QCD is the result of the highly nonlinear nature of strong interactions and a large running coupling at low probe energies on the order of 1 GeV, because this seems to foreclose the small-coupling perturbative treatments which are effective, for example, in QED with its small running "fine structure" coupling ~1/137.036 and inverse-square force. In the results here, however, the non-linearities of QCD including the jets required to maintain color neutrality and color confinement are fully embedded in the infinitely recursive nature of (24), which provides an exact analytic expression for the monopole baryon in closed recursive form. So to carry through the analytics, one would resubstitute the gauge fields (21) into (24) in the manner of (22) over and over. Even more preferably, one would try to discern in closed from, the infinite series that emerges from these recursive resubstitutions, because this would finally produce the analytic QCD solution that lattice QCD only achieves numerically.

So, returning to Rabi, who ordered the baryons? Fundamentally, because a baryon is described by Maxwell's equation for a magnetic monopole in covariant form, populated by fermions housed in Maxwell's inverted equation for an electric charge, all generalized to non-abelian SU(N) Yang–Mills gauge theory, the answer is that they were primarily ordered by Maxwell and Yang and Mills, whose theories have gained universal confidence as to their underlying validity. Moreover, the similarly-established surface flux laws first developed by Gauss play a crucial role, because they specify both what is confined within closed YM monopole surfaces (net color thus colored quarks and bi-colored gluons) and what is allowed to net flow across these surfaces (colorless quark-antiquark pairs including colorless mesons and jets). A role is also assumed by Dirac whose quantum theory of the electron first exposited the existence and nature of fermions, and by Weyl for elaborating gauge theory itself. That these monopoles contain exactly three fermions in their ground state which fit the fundamental representation of SU(3) also uses the Exclusion Principle of Fermi, Dirac and Pauli. This "three-ness" itself, as well as the color-neutral nature of a baryon, is rooted in the third-rank antisymmetric tensor first taught by Einstein for the generally covariant description of magnetic monopoles. Finally, one should not overlook Hamilton who first developed non-commuting quaternions in order to compactly describe simple rotations in three space dimensions. This is because non-commuting mathematical objects subsequently came to play a central role in many areas of physics, including the extension of quaternions and their Pauli spin matrix descendants into the non-commuting vector potentials of Yang–Mills gauge theory. Importantly, all of these theories and theoretical tools are well-settled, uncontradicted and universally accepted. All that is novel in this present work, is the illumination of how these all combine to produce the observed QCD physics of hadrons.

## 17. Filling the Yang–Mills Mass Gap

It is well-known that Maxwell's electric charge Equation (1a) has no inverse, or, to be precise, that the inverse $(g^{\mu\nu}\partial_\sigma\partial^\sigma - \partial^\mu\partial^\nu)^{-1}$ of its operator on $A_\mu$ is infinite (singular). To deal with this, one of several approaches is typically required. A first option is to impose a gauge condition, often the covariant $\partial_\mu A^\mu = 0$. Then it is easy to obtain $(g^{\mu\nu}\partial_\sigma\partial^\sigma)^{-1}$ alone. Another is to introduce a Proca mass by hand, which is what we did at (15). However, doing so means the theory is no longer renormalizable, so we must eventually find a way to remove this mass and introduce it some other way. Using $D \mapsto \partial$ to obtain a "signal" inverse, the inverse of the operator in (15) becomes the familiar finite (A8). We also see from (A8) that when $m = 0$ this inverse becomes infinite, which directly demonstrates why (1a) has no inverse. Using notations reviewed at the start of Section 6, this is not just because of $k_\nu k_\alpha / m^2$ in the numerator which can be removed when contracted with a source current because of the signal continuity equation $k_\nu j^\nu = 0$ reviewed near the end of Appendix B. More importantly, it is because $k_\sigma k^\sigma - m^2$ in the denominator becomes zero when the vector boson is "on-shell," meaning that $k_\sigma k^\sigma - m^2 = 0$. When we remove the Proca mass so the boson is again massless, the denominator becomes $q_\sigma q^\sigma$ which on-shell is also $q_\sigma q^\sigma = 0$. These singularities are why

we also need the $+i\varepsilon$ prescription to take a particle "off-shell," i.e., to render it "virtual" about the singular "pole."

In view of this, we sum the inverse (20) from the left with electric source density $J^\alpha$ and from the right with $J^\nu$, both from (10a). Recall, using $c^2\mu_0\varepsilon_0 = 1$, that $J^\sigma = j^\sigma - ic\varepsilon_0 g G_\tau F^{\tau\sigma}$ as reviewed following (12). As obtained in (A15) the continuity equation $(p_\nu + gG_\nu)J^\nu = 0$ whereby the term $(k_\nu k_\alpha + gk_\nu(G_\alpha)/c)/m^2c^2 = 0$ and so drops out. So, with $c = 1$ we obtain:

$$J^\alpha I_{\alpha\nu}J^\nu = J^\alpha \frac{\vee\left(-g_{\alpha\nu} + \frac{k_\nu k_\alpha + gk_\nu G_\alpha}{m^2}\right)}{''k_\sigma k^\sigma - m^2 - g^2 G_\sigma G^\sigma + i\varepsilon''} J^\nu = -J_\sigma\left(k_\sigma k^\sigma - m^2 - g^2 G_\sigma G^\sigma + i\varepsilon\right)^{-1} J^\sigma \tag{45}$$

Now, we see how to remove the Proca mass thus restoring renormalizability while maintaining a finite inverse: Because the term $g^2 G_\sigma G^\sigma$ arises naturally from the Yang–Mills gauge theory and is itself square-mass-dimensioned, and because this term has the same form as what is in $\mathcal{L} = \partial^\sigma\phi * \partial_\sigma\phi + g^2 G^\sigma G_\sigma\phi * \phi$ whereby vector boson masses arise in a well-known way from spontaneous symmetry breaking, we simply use $g^2 G_\sigma G^\sigma$ to replace $m^2$ by removing the latter entirely. In other words, Yang–Mills gauge theory puts a mass-producing term right where it needs to be in the form it needs to have, so we can set the mass added by hand at (15) back to $m = 0$ in the above without adverse consequence. Likewise, neither do we need $+i\varepsilon$. So, removing these, and expanding $G_\sigma = \tau_i G_{i\sigma}$, (45) now becomes:

$$J^\alpha I_{\alpha\nu}J^\nu = -J_\sigma\left(k_\sigma k^\sigma - g^2 G_\sigma G^\sigma\right)^{-1} J^\sigma = -J_\sigma\left(k_\sigma k^\sigma - g^2\tau_i\tau_j g_{\mu\nu} G_i{}^\mu G_j{}^\nu\right)^{-1} J^\sigma \tag{46}$$

What is important now is that $G_\sigma G^\sigma = \tau_i\tau_j g_{\mu\nu} G_i{}^\mu G_j{}^\nu$ is not an ordinary spacetime scalar, but rather, is an N×N matrix for any Yang–Mills gauge group SU(N), with adjoint structure established by the $\left(N^2 - 1\right)^2$ products $\tau_i\tau_j$. Furthermore, some of the matrices $\tau_i\tau_j$ and thus $g_{\mu\nu} G^\mu G^\nu$ contain imaginary components, which can be seen even in the simplest case of SU(2) where $\tau_i = \frac{1}{2}\sigma_i$ and the Pauli identity $(\boldsymbol{\sigma}\cdot\mathbf{x})(\boldsymbol{\sigma}\cdot\mathbf{y}) = \mathrm{Id}\,\mathbf{x}\cdot\mathbf{y} + i\boldsymbol{\sigma}\cdot(\mathbf{x}\times\mathbf{y})$. The $J^\sigma = j^\sigma - ic\varepsilon_0 g G_\tau F^{\tau\sigma}$ which are also N×N YM matrices will contain additional imaginary components. Moreover, nothing restricts (46) to SU(3). As reviewed in Section 10, this restriction is imposed by the magnetic monopoles. Here, we are dealing with Maxwell's Yang–Mills Equation (12a) for electric source densities independent of magnetic monopoles, with the Proca mass of (15) now removed.

Because (46) is an N×N Yang–Mills matrix and a spacetime scalar, and additionally houses a finite-matrix ($f$) inverse which by definition is invertible, i.e., $\left(f^{-1}\right)^{-1} = f$, it has finite eigenvalues $\lambda$ which are calculated in the usual way for a square matrix $M$ via the determinant relation $0 = |M - \lambda\,\mathrm{Id}|$, where Id is an N×N unit matrix for any SU(N). These $\lambda$ are obtained by:

$$0 = \left|J^\alpha I_{\alpha\nu}J^\nu - \lambda\,\mathrm{Id}\right| = \left|-J_\sigma\left(k_\tau k^\tau - g^2 G_\tau G^\tau\right)^{-1} J^\sigma - \lambda\,\mathrm{Id}\right| \tag{47}$$

However, (A8) provides the basis for seeing what these eigenvalues look like in the non-perturbative limit with all recursion reviewed at (21) and (22) removed via $D_\sigma \mapsto \partial_\sigma$ so that $c\mu_0 J^\nu = D_\sigma F^{\sigma\nu}$ becomes $c\mu_0 j^\nu = \partial_\sigma F^{\sigma\nu}$ and $J^\sigma$ becomes $j^\sigma$. In Section 6 notation with what is now a continuity equation $k_\nu j^\nu = 0$ from Appendix B, these eigenvalues become:

$$\lambda = j^\alpha i_{\alpha\nu}j^\nu = j^\alpha \frac{-g_{\alpha\nu} + \frac{k_\nu k_\alpha}{m^2}}{k_\sigma k^\sigma - m^2 + i\varepsilon} j^\nu = \frac{-j_\sigma j^\sigma}{k_\sigma k^\sigma - m^2 + i\varepsilon} \tag{48}$$

Finally, combining (47) and (48) and reintroducing the infinite recursive series notation of (22) with … wherever a substitution $G_\alpha(j^\nu, G_\alpha)$ or $F^{\mu\nu}(G_\alpha(j^\nu, G_\alpha))$ is required, and noting the similarity of

$J^\alpha I_{\alpha\nu}J^\nu - j^\alpha i_{\alpha\nu}j^\nu\text{Id}$ to $-V = D_\sigma D^\sigma - \partial_\sigma\partial^\sigma$ discussed at the start of Section 4 whereby the perturbation is the difference "noise = (signal + noise) − signal," we obtain:

$$
\begin{aligned}
0 = \left|J^\alpha I_{\alpha\nu}J^\nu - j^\alpha i_{\alpha\nu}j^\nu\text{Id}\right| &= \left|-J_\sigma\big(k\,k^\tau - g^2(G_\tau\ldots)(G^\tau\ldots)\big)^{-1}J^\sigma - \frac{-j_\sigma j^\sigma}{k_\sigma k^\sigma - m^2 + i\varepsilon}\text{Id}\right| \\
&= \left|-(j_\sigma - ic\varepsilon_0 g(G^\alpha F_{\alpha\sigma}\ldots))\big(k_\tau k^\tau - g^2(G_\tau\ldots)(G^\tau\ldots)\big)^{-1}\big(j^\sigma - ic\varepsilon_0 g(G_\beta F^{\beta\sigma}\ldots)\big) - \frac{-j_\sigma j^\sigma}{k_\sigma k^\sigma - m^2 + i\varepsilon}\text{Id}\right|
\end{aligned}
\tag{49}
$$

*This is the mass gap solution.* Specifically, referring to page 6 of [2], for "any compact simple gauge group G," that is, for any SU(N), "a non-trivial quantum Yang–Mills theory exists on $\mathbb{R}^4$ and has a mass gap > 0 namely there must be some constant $\Delta > 0$ such that every excitation of the vacuum has energy at least $\Delta$". "Excitations of the vacuum" is another phrase for non-zero perturbations $V_{\nu\sigma} = ig(G_\nu\partial_\sigma + \partial_\nu G_\sigma) + g^2 G_\nu G_\sigma \neq 0$ defined following (11), and these arise from the canonic promotion of $\partial_\sigma \mapsto D_\sigma$ reviewed in Section 3. It is also another phrase for what we have referred to here as the "noise" of Yang–Mills dynamics over Maxwell's U(1) electrodynamics. Indeed, $J^\alpha I_{\alpha\nu}J^\nu - j^\alpha i_{\alpha\nu}j^\nu\text{Id}$ in the determinant (49), like $V$, is also a direct measure of "noise = (signal + noise) − signal." When these excitations of the vacuum $V_{\nu\sigma} \neq 0$, the inverse $G_\alpha(j^\nu, G_\alpha(j^\nu, G_\alpha(j^\nu, G_\alpha(j^\nu, G_\alpha(\ldots)))))$ has the infinitely recursive, highly nonlinear form of (22). However, $J_\sigma\big(k_\tau k^\tau - g^2 G_\tau G^\tau\big)^{-1}J^\sigma$ is clearly invertible (again, $\left(f^{-1}\right)^{-1} = f$ ), so it will have a non-zero determinant and non-zero eigenvalues.

Further, $G_\sigma G^\sigma$ is a correctly signed term for positive vector boson rest energy $mc^2 > 0$, has the same form as what appears in the symmetry-breaking $\mathcal{L} = \partial^\sigma\phi * \partial_\sigma\phi + g^2 G^\sigma G_\sigma\phi * \phi$, and at (46) was used to remove the Proca mass to restore renormalizability while maintaining finite invertibility. Because $-J_\sigma\big(k_\tau k^\tau - g^2(G_\tau\ldots)(G^\tau\ldots)\big)^{-1}J^\sigma$ contains both real and imaginary numbers, these eigenvalues can be real, imaginary, or complex. So, when the eigenvalues (48) and then the $-m^2 + i\varepsilon$ in the eigenvalue denominator are computed using (49) for "*any* compact simple gauge group G," we will find that "every excitation of the vacuum has energy at least $\Delta$ "= $mc^2 > 0$. Moreover, the imaginary parts of $J^\alpha I_{\alpha\nu}J^\nu$ in (49) can produce $+i\varepsilon > i0$ which corresponds physically to finite particle lifetimes. This is how Yang–Mills gauge theory reveals masses $m > 0$ and finite lifetimes via $\varepsilon > 0$ in the (48) denominator while maintaining renormalizability by removing Proca masses and $+i\varepsilon$, thereby filling the Yang–Mills mass gap.

The manifest recursion highlighted in (49) is also important for understanding how to carry out exact closed analytic calculations in Yang–Mills theory, as opposed to using numerical methods such as those of Lattice QCD. In this regard, Jaffe and Witten state on page 7 of [2] that "since the inception of quantum field theory, two central methods have emerged to show the existence of quantum fields on non-compact configuration space (such as Minkowski space). These known methods are (i) Find an exact solution in closed form; (ii) Solve a sequence of approximate problems, and establish convergence of these solutions to the desired limit." The foregoing (49) suggests a third method which is really a hybrid of (i) and (ii): find an exact recursive kernel in closed form, and then expand that kernel in successive iterations approaching the limit of infinite recursive nesting to identify the underlying infinite series in a closed form. As noted toward the end of Section 16, in the specific context of QCD rather than the general context of any compact simple Yang–Mills gauge group, this is the basis for developing an exact, closed analytical form of QCD, versus having to resort only to the numerical methods of lattice QCD.

## 18. Conclusions

For an entire century we have known experimentally about the existence of protons. For almost 90 years we have known about neutrons. However, beyond knowing baryons contain three quarks with exact SU(3) chromodynamic symmetry, contain massless gluons in the adjoint representation of SU(3), have an antisymmetric color-neutral singlet wavefunction, and interact via symmetric color-neutral

singlet mesons, we still cannot answer Rabi's simple query "who ordered that?," and we do not understand the dynamic basis of quark and gluon confinement and hadronization.

Here, we have shown that the magnetic monopoles of Yang–Mills gauge theory in their "signal" state contain three fermions in the fundamental representation of SU(3). Following symmetry breaking which moves a degree of freedom from the gauge bosons to the fermions, the gauge bosons become massless, SU(3) becomes an exact symmetry, and a propagator is established for each fermion. The monopoles then have the same antisymmetric color singlet wavefunction as a baryon, and the field quanta of the magnetic fields fluxing through their surface have the same symmetric color singlet wavefunction as a meson. Consequently, we can identify these fermions with colored quarks, the massless gauge bosons with gluons, the magnetic monopoles with baryons, the fluxing entities with mesons, and the symmetry breaking with hadronization, while establishing that the quarks and gluons remain confined following hadronization. The result is a quantum chromodynamic (QCD) theory of the hadrons. Using analytic tools developed along the way, we also fill the Yang–Mills mass gap.

Finally, as previewed in the introduction and further detailed in Section 16, Rabi's question is answered: Protons, neutrons and other baryons were ordered primarily by Maxwell, Gauss, Yang and Mills. Additional foundations were provided by Weyl who was the father of gauge theory, as well as Fermi, Dirac and Pauli via Dirac's quantum theory of the electron and the fermion Exclusion Principle, and Einstein's generally covariant formulation of Maxwell's magnetic monopoles as a third rank antisymmetric tensor which directly accounts for ground state baryons containing three colored quarks while remaining color-neutral. Finally, seminal credit must be attributed to Hamilton for pioneering non-commuting quaternions which more than a century later became the foundation of Yang–Mills gauge theory.

**Funding:** This research received no external funding.

**Conflicts of Interest:** The author declares no conflict of interest.

## Appendix A. Calculation of the Inverse for the Yang–Mills Electric Source Equation

Because $\partial^\mu D^\nu = \partial^\mu \partial^\nu - ig\partial^\mu G^\nu$ is a non-symmetric tensor even in flat spacetime because in general $\partial^\mu G^\nu \neq \partial^\nu G^\mu$, it is important when calculating the inverse $I_{\alpha\nu}$ to make certain that the left- and right-side inverse calculations lead to the same $\delta^\mu{}_\alpha$ identity matrix, with $I_{\alpha\nu\text{LEFT}} = I_{\alpha\nu\text{RIGHT}}$, as shown at (18). Therefore, we shall carry out both a left- and a right-side calculation, then make certain that both of these inverses are one and the same. Based on (18), we expect the general form of the inverse to be:

$$I_{\alpha\nu\text{LEFT}} = I_{\alpha\nu\text{RIGHT}} = Ag_{\alpha\nu} + B\partial_\alpha D_\nu + C\partial_\nu D_\alpha \tag{A1}$$

where $A$, $B$ and $C$ are unknowns to be determined, and where we include both $\partial_\alpha D_\nu$ and $\partial_\nu D_\alpha$ given the non-symmetry of these terms. The left-placement of $A$, $B$ and $C$ in (A1) is arbitrary ab initio, but once we do so, we must maintain consistent ordering thereafter. So, it would be incorrect to write $I_{\alpha\nu\text{RIGHT}} = g_{\alpha\nu}A + \partial_\alpha D_\nu B + \partial_\nu D_\alpha C$. It should also be noted from the term $\partial^\mu D^\nu G_\mu$ in (15) that the free index is in $D^\nu$, while $\partial^\mu$ to the left of $D^\nu$ sums with $G_\mu$. We will calculate in configuration space, then convert to momentum space in the usual way.

Using (A1) in (18) as a left-side inverse and operating with the metric tensor produces:

$$\begin{aligned}
\delta^\mu{}_\alpha &= (Ag_{\alpha\nu} + B\partial_\alpha D_\nu + C\partial_\nu D_\alpha)\big(g^{\mu\nu}\big(\partial_\sigma D^\sigma + m^2\big) - \partial^\mu D^\nu\big) \\
&= A\delta^\mu{}_\alpha\big(\partial_\sigma D^\sigma + m^2\big) - A\partial^\mu D_\alpha + B\partial_\alpha D^\mu\big(\partial_\sigma D^\sigma + m^2\big) - B\partial_\alpha D_\sigma \partial^\mu D^\sigma \\
&\quad + C\partial^\mu D_\alpha\big(\partial_\sigma D^\sigma + m^2\big) - C\partial_\sigma D_\alpha \partial^\mu D^\sigma
\end{aligned} \tag{A2}$$

Matching up $\delta^\mu{}_\alpha$ with the term $A\delta^\mu{}_\alpha \partial_\sigma D^\sigma$ first reveals that $\delta^\mu{}_\alpha = A\delta^\mu{}_\alpha\big(\partial_\sigma D^\sigma + m^2\big)$, i.e., that:

$$A = \big(\partial_\sigma D^\sigma + m^2\big)^{-1} \tag{A3}$$

Because $D^\sigma = \partial^\sigma - igA^\sigma$ contains $A^\sigma = \tau_i A_i^\sigma$ which is an N×N square matrix for SU(N), we cannot simply write the above as $A = 1/(\partial_\sigma D^\sigma + m^2)$ which treats $\partial_\sigma D^\sigma + m^2$ as an ordinary denominator. Rather, this must itself be inverted independently of the spacetime inversion (18).

Substituting (A3) into (A2) then reducing now produces:

$$
\begin{aligned}
&\left(\partial_\sigma D^\sigma + m^2\right)^{-1} \partial^\mu D_\alpha \\
&= B\left(\partial_\alpha D^\mu \left(\partial_\sigma D^\sigma + m^2\right) - \partial_\alpha D_\sigma \partial^\mu D^\sigma\right) + C\left(\partial^\mu D_\alpha \left(\partial_\sigma D^\sigma + m^2\right) - \partial_\sigma D_\alpha \partial^\mu D^\sigma\right)
\end{aligned}
\tag{A4}
$$

The left side above contains $\partial^\mu D_\alpha$ which matches to the same term inside $C\partial^\mu D_\alpha\left(\partial_\sigma D^\sigma + m^2\right)$. From this we conclude that the terms with $B$ are not needed to calculate the inverse, i.e., that we can calculate the inverse with $B = 0$. This is a downstream consequence of the fact noted following (A1) that in (15), the free index is in $D^\nu$. Consequently, we further reduce (A4) to:

$$
C = \left(\partial_\sigma D^\sigma + m^2\right)^{-1}\left[\partial^\mu D_\alpha \left(\partial^\mu D_\alpha m^2 + \partial^\mu D_\alpha \partial_\sigma D^\sigma - \partial_\sigma D_\alpha \partial^\mu D^\sigma\right)^{-1}\right]
\tag{A5}
$$

Inserting (A3) and $B = 0$ and (A5) into (A1) and reducing now produces the left-side inverse:

$$
\begin{aligned}
I_{\alpha\nu\text{LEFT}} &= \left(\partial_\sigma D^\sigma + m^2\right)^{-1}\left\{g_{\alpha\nu} + \partial^\mu D_\alpha \left(\partial^\mu D_\alpha m^2 + \partial^\mu D_\alpha \partial_\sigma D^\sigma - \partial_\sigma D_\alpha \partial^\mu D^\sigma\right)^{-1}\partial_\nu D_\alpha\right\} \\
&\equiv \frac{\vee\left(g_{\alpha\nu} + \dfrac{\partial^\mu D_{\alpha\vee}\partial_\nu D_\alpha}{"m^2\partial^\mu D_\alpha + \partial^\mu D_\alpha\partial_\sigma D^\sigma - \partial_\sigma D_\alpha\partial^\mu D^\sigma"}\right)}{"\partial_\sigma D^\sigma + m^2"}
\end{aligned}
\tag{A6}
$$

In the bottom line above, simply to provide a compact visual comparison to the usual inverses for a massive vector boson, we have defined "quoted" denominators in which the inverses are represented as "denominators", but with the understanding that this is a shorthand for what is actually a matrix inverse. In general, for a square matrix $M$, we shall use this shorthand to write $1/''M'' \equiv M^{-1}$. We also use a subscripted $\vee$ to indicate where the "denominators" are placed when represented as inverses. Looking closely, note that $\partial^\mu D_\alpha$ appears in both the upper numerator and the upper "denominator," but cannot (yet) be cancelled using $(\partial^\mu D_\alpha)(\partial^\mu D_\alpha)^{-1} = \text{Id}$ because of the $\partial_\sigma D_\alpha \partial^\mu D^\sigma$ term in the upper denominator in which $D_\alpha$ is commuted to the left of $\partial^\mu$.

Next, we perform an identical calculation using (18), again using the general form (A1), but now for $I_{\alpha\nu\text{RIGHT}}$. The result for $A$ is the same as in (A3), and by matching up the $\partial^\mu D_\alpha$ terms as in (A4) we again conclude that $B = 0$. So, we finally calculate $C$ as in (A5) and insert all the results into (A1) to find that:

$$
\begin{aligned}
I_{\alpha\nu\text{RIGHT}} &= \left(\partial_\sigma D^\sigma + m^2\right)^{-1}\left\{g_{\alpha\nu} + \partial^\mu D_\alpha \left(\partial^\mu D_\alpha m^2 + \partial_\sigma D^\sigma \partial^\mu D_\alpha - \partial^\mu D^\sigma \partial_\sigma D_\alpha\right)^{-1}\partial_\nu D_\alpha\right\} \\
&\equiv \frac{\vee\left(g_{\alpha\nu} + \dfrac{\partial^\mu D_{\alpha\vee}\partial_\nu D_\alpha}{"\partial^\mu D_\alpha m^2 + \partial_\sigma D^\sigma \partial^\mu D_\alpha - \partial^\mu D^\sigma \partial_\sigma D_\alpha"}\right)}{"\partial_\sigma D^\sigma + m^2"}
\end{aligned}
\tag{A7}
$$

If we set $D \mapsto \partial$ throughout to turn the gauge-covariant derivatives into ordinary ones, then use $\left[\partial_\alpha, \partial_\beta\right] = 0$ in flat spacetime, and then convert from configuration into momentum space using the substitution $ih\partial^\mu \mapsto p^\mu$ with $h = 1$ while including $+i\varepsilon$, the quotes can come off the denominators, and we find that the non-perturbative $I_{\alpha\nu} \mapsto i_{\alpha\nu}$ inverse is:

$$
i_{\alpha\nu} \equiv \frac{g_{\alpha\nu} + \dfrac{\partial_\nu \partial_\alpha}{m^2}}{\partial_\sigma \partial^\sigma + m^2 - i\varepsilon} = \frac{-g_{\alpha\nu} + \dfrac{p_\nu p_\alpha}{m^2}}{p_\sigma p^\sigma - m^2 + i\varepsilon}
\tag{A8}
$$

This will be recognized as the well-known inverse for a massive vector boson. Using the language of Jaffe and Witten in [2], this has no "excitations of the vacuum," and is used in the eigenvalues (48) for the mass gap solution (49).

Now, the requirement $M^{-1}M = MM^{-1} = \text{Id}$ for any square matrix $M$ tells us that the inverse $M^{-1}$ must be the same no matter the side from which it multiplies $M$, that is, $I_{\alpha\nu\text{LEFT}} = I_{\alpha\nu\text{RIGHT}}$. So, if we now set the two results in (A6) and (A7) to be equal, we find this will be so if and only if:

$$\partial^\mu D_\alpha \partial_\sigma D^\sigma - \partial_\sigma D_\alpha \partial^\mu D^\sigma = \partial_\sigma D^\sigma \partial^\mu D_\alpha - \partial^\mu D^\sigma \partial_\sigma D_\alpha (= 0) \tag{A9}$$

As it turns out, not only are both sides of (A9) equal as required, but each is equal to zero which is why we included $(= 0)$. This can be proved using the covariant commutator relation $\left[p_\mu, G_\nu\right] = -ih\partial_\mu G_\nu$ for a field $G_\nu(t, \mathbf{x})$ which is a function of space and time. The $\left[p_i, G_j\right] = -ih\partial_i G_j$ space components originate in the Heisenberg commutator $[\hat{p}_x, \hat{x}] = -ih$, while the time component is rooted in the Heisenberg-picture commutator $\left[\hat{H}, G_\nu\right] = -ih\partial_0 G_\nu$ in view of the relation $\left\langle \overline{\psi}|\hat{H}|\psi\right\rangle = \left\langle \overline{\psi}|cp^0|\psi\right\rangle$, where $\hat{H}$ is a particle Hamiltonian and $cp^0 = E$ is the particle energy.

We can see this in the following way: Substitute $D^\sigma = \partial^\sigma - igG^\sigma$ throughout (A9). Then factor out one $g$, and use $\left[\partial_\alpha, \partial_\beta\right] = 0$ in flat spacetime to remove some terms, yielding:

$$
\begin{aligned}
& i\partial_\sigma G_\alpha \partial^\mu \partial^\sigma - i\partial^\mu G_\alpha \partial_\sigma \partial^\sigma + g\partial_\sigma G_\alpha \partial^\mu G^\sigma - g\partial^\mu G_\alpha \partial_\sigma G^\sigma \\
& = i\partial^\mu G^\sigma \partial_\sigma \partial_\alpha - i\partial_\sigma G^\sigma \partial^\mu \partial_\alpha + g\partial^\mu G^\sigma \partial_\sigma G_\alpha - g\partial_\sigma G^\sigma \partial^\mu G_\alpha
\end{aligned}
\tag{A10}
$$

Next, convert the partial derivative which is just to the right of a gauge field in each of the eight terms, into momentum space using $ih\partial^\mu \mapsto p^\mu$ with $h = 1$. Then use $\left[p_\mu, G_\nu\right] = -i\partial_\mu G_\nu$ to commute these new $p^\mu$ to the left of the gauge fields. More terms cancel using $\left[\partial_\alpha, \partial_\beta\right] = 0$, so:

$$
\begin{aligned}
& \partial_\sigma p^\mu G_\alpha \partial^\sigma - \partial^\mu p_\sigma G_\alpha \partial^\sigma + ig\partial^\mu p_\sigma G_\alpha G^\sigma - ig\partial_\sigma p^\mu G_\alpha G^\sigma \\
& = \partial^\mu p_\sigma G^\sigma \partial_\alpha - \partial_\sigma p^\mu G^\sigma \partial_\alpha + ig\partial_\sigma p^\mu G^\sigma G_\alpha - ig\partial^\mu p_\sigma G^\sigma G_\alpha
\end{aligned}
\tag{A11}
$$

Finally, revert to configuration space via $p^\mu \mapsto i\partial^\mu$, make one final use of $\left[\partial_\alpha, \partial_\beta\right] = 0$, then multiply through by $-i$ and consolidate terms using $D_\mu = \partial_\mu - igG_\mu$ to obtain:

$$
\begin{aligned}
& (\partial_\sigma \partial^\mu - \partial^\mu \partial_\sigma) G_\alpha (\partial^\sigma - igG^\sigma) = [\partial_\sigma, \partial^\mu] G_\alpha D^\sigma = 0 \\
& = (\partial^\mu \partial_\sigma - \partial_\sigma \partial^\mu) G^\sigma (\partial_\alpha - igG_\alpha) = [\partial^\mu, \partial_\sigma] G^\sigma D_\alpha = 0
\end{aligned}
\tag{A12}
$$

The result is simply $0 = 0$ with both the left and right sides seen to equal zero.

This has two important consequences: First, it proves that (A9) is not some independent condition on the gauge fields, but is simply a manifestation of the commutator relation $\left[p_\mu, G_\nu\right] = -ih\partial_\mu G_\nu$ which covariantly combines both the canonical commutation relations in space and the Heisenberg equation of motion commutator in time. Second, because both sides of (A12) are not only equal, but are each independently equal to zero, we prove that each side of (A9) is independently equal to zero in view of the covariant commutator $\left[p_\mu, G_\nu\right] = -ih\partial_\mu G_\nu$. A third logical consequence is that unless there is some other way to go from (A9) from (A12) without using $\left[p_\mu, G_\nu\right] = -ih\partial_\mu G_\nu$ (which may be possible but is not apparent), then $\left[p_\mu, G_\nu\right] = -ih\partial_\mu G_\nu$ is retro-proved by the linear algebra requirement that $M^{-1}M = MM^{-1} = \text{Id}$ for a square matrix inverse, in the current context which includes the Heisenberg picture.

Because each side of (A9) is equal to zero, we may set these same terms to zero in each of (A6) and (A7), which are then clearly equal to one another, $I_{\alpha\nu\text{LEFT}} = I_{\alpha\nu\text{RIGHT}}$. Then we can reduce each of (A6) and (A7) via $(\partial^\mu D_\alpha)(\partial^\mu D_\alpha)^{-1} = \text{Id}$, then apply $D^\sigma = \partial^\sigma - igG^\sigma$, then apply $\partial_\sigma G^\sigma = igG_\sigma G^\sigma$

from (16) which, again, is a required condition for a massive vector boson in Yang–Mills gauge theory, then apply a final $i\partial^\mu \mapsto p^\mu$ conversion to momentum space and add the $+i\varepsilon$ prescription, to obtain:

$$
\begin{aligned}
I_{\alpha\nu} &= \left(\partial_\sigma\partial^\sigma + m^2 + g^2 G_\sigma G^\sigma + i\varepsilon\right)^{-1}\left(g_{\alpha\nu} + \frac{\partial_\nu\partial_\alpha - ig\partial_\nu G_\alpha}{m^2}\right) \\
&= \left(-p_\sigma p^\sigma + m^2 + g^2 G_\sigma G^\sigma + i\varepsilon\right)^{-1}\left(g_{\alpha\nu} - \frac{p_\nu p_\alpha + g p_\nu G_\alpha}{m^2}\right) = \frac{\vee\left(-g_{\alpha\nu} + \frac{p_\nu p_\alpha + g p_\nu G_\alpha}{m^2}\right)}{''p_\sigma p^\sigma - m^2 - g^2 G_\sigma G^\sigma + i\varepsilon''}
\end{aligned}
\tag{A13}
$$

It is easily seen that when $p_\nu G_\alpha = 0$ and $G_\sigma G^\sigma = 0$, this reduces to the standard inverse (A8) for a massive vector boson. Conversely, this means the two terms $g p_\nu G_\alpha$ in the numerator and $g^2 G_\sigma G^\sigma$ in the inverse "denominator" are what get added to the massive vector boson inverse by Yang–Mills gauge theory perturbations. Furthermore, as noted at (16) and used at (46), $g^2 G_\sigma G^\sigma$ is precisely the term in which boson masses are revealed via $\mathcal{L} = \partial^\sigma\phi * \partial_\sigma\phi + g^2 G^\sigma G_\sigma\phi * \phi$ during the spontaneous symmetry breaking of renormalizable gauge theory.

## Appendix B. The Yang–Mills Continuity Equation in Terms of Dirac Wavefunctions

To obtain a Yang–Mills continuity equation in terms of Dirac wavefunctions, we start with Dirac's Yang–Mills canonic equation $i\gamma^\sigma D_\sigma\psi - m\psi = 0$. With the adjoint wavefunction $\overline{\psi} \equiv \psi^\dagger\gamma^0$ defined as usual, it is straightforward to obtain the adjoint equation $iD_\sigma{}^\dagger\overline{\psi}\gamma^\sigma + m\overline{\psi} = 0$. If we then sandwich each of these and add, because the $igG_\sigma$ terms from $D_\sigma$ and $D_\sigma{}^\dagger$ cancel out, we obtain:

$$
\begin{aligned}
0 &= \overline{\psi}\gamma^\sigma D_\sigma\psi + D_\sigma{}^\dagger\overline{\psi}\gamma^\sigma\psi = \overline{\psi}\gamma^\sigma(\partial_\sigma - igG_\sigma)\psi + (\partial_\sigma + igG_\sigma)\overline{\psi}\gamma^\sigma\psi \\
&= \overline{\psi}\gamma^\sigma\partial_\sigma\psi + \partial_\sigma\overline{\psi}\gamma^\sigma\psi = \partial_\sigma\left(\overline{\psi}\gamma^\sigma\psi\right) = 0
\end{aligned}
\tag{A14}
$$

So, even for Yang–Mills theory, the continuity relation for Dirac wavefunctions only contains the ordinary derivative.

Combining (A14) with (11) this also means, matching up continuity zeros, that:

$$
\begin{aligned}
0 &= c\mu_0 D_\nu J^\nu = D_\nu D_\sigma F^{\sigma\nu} = \partial_\nu\partial_\sigma F^{\sigma\nu} - V_{\nu\sigma}F^{\sigma\nu} = c\mu_0\partial_\sigma\left(g\overline{\psi}\gamma^\sigma\psi\right) \\
&= c\mu_0(\partial_\nu - igG_\nu)J^\nu = -ic\mu_0(p_\nu + gG_\nu)J^\nu
\end{aligned}
\tag{A15}
$$

which via $i\partial_\mu \mapsto p_\mu$ contains the continuity relation $(p_\nu + gG_\nu)J^\nu = 0$ mentioned after (11) and used in (45). Combining this with $c\mu_0 j^\nu = \partial_\sigma F^{\sigma\nu}$ from (12a) and using (11), we then obtain:

$$
c\mu_0\partial_\nu j^\nu = \partial_\nu\partial_\sigma F^{\sigma\nu} = c\mu_0 D_\nu J^\nu + V_{\nu\sigma}F^{\sigma\nu} = c\mu_0\partial_\nu\left(g\overline{\psi}\gamma^\nu\psi\right) + V_{\nu\sigma}F^{\sigma\nu} = V_{\nu\sigma}F^{\sigma\nu}
\tag{A16}
$$

including the perturbation tensor $V_{\nu\sigma} = ig(G_\nu\partial_\sigma + \partial_\nu G_\sigma) + g^2 G_\nu G_\sigma$ defined at (11). When $V_{\nu\sigma} = 0$ this reduces following integration without integration constant to the familiar $j^\nu = g\overline{\psi}\gamma^\nu\psi$.

Finally, because of the perturbations, i.e., excitations of the vacuum of Yang–Mills theory, it is beneficial to define a four-vector $\kappa^\nu$ with dimensions of charge density, which is also an N×N matrix for SU(N), such that $j^\nu \equiv g\overline{\psi}\gamma^\nu\psi + \kappa^\nu$. Then, because (A14) reveals $\partial_\nu\left(g\overline{\psi}\gamma^\nu\psi\right) = 0$ even in Yang–Mills theory, we can insert this definition into (A16) to deduce $c\mu_0\partial_\nu\kappa^\nu = V_{\nu\sigma}F^{\sigma\nu}$. Using $c^2\mu_0\varepsilon_0 = 1$, and mindful that $c\mu_0 J^\nu = c\mu_0 j^\nu - igG_\sigma F^{\sigma\nu} = D_\sigma F^{\sigma\nu}$, the net result is that:

$$
j^\nu = g\overline{\psi}\gamma^\nu\psi + \kappa^\nu
\tag{A17}
$$

$$
\partial_\nu\kappa^\nu = c\varepsilon_0 V_{\nu\sigma}F^{\sigma\nu}
\tag{A18}
$$

with the latter being a scalar first-order differential equation for $\kappa^\nu$ When the perturbation tensor $V_{\nu\sigma} = 0$ (A16) reduces to $\partial_\nu j^\nu = \partial_\nu\left(g\overline{\psi}\gamma^\nu\psi\right) = 0$, so that with integration constants set to zero we recover the familiar $j^\nu = \overline{\psi}\gamma^\nu\psi$ with $\kappa^\nu = 0$. Likewise when $V_{\nu\sigma} = 0$ the continuity equation (A16) becomes $\partial_\nu j^\nu = 0$, which in momentum space via $i\partial^\mu \mapsto p^\mu$ further means that $p_\nu j^\nu = 0$.

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
