# Peer review of "QCD Theory of the Hadrons and Filling the Yang–Mills Mass Gap"

_symmetry, doi:10.3390/sym12111887_

Round 1

Reviewer 1 Report

Dear Authors,

The paper with title "QCD theory of the hadrons and filling the Yang-
3 Mills mass gap" is good written and very interesting. In my opinion paper also could be very important for scientific reader. I only recomend authors if possible extend discussion part of paper and compare or discuss how their approach could influent for example recent results published in papers :

International Journal of Geometric Methods in Modern Physics
Vol. 14, No. 9 (2017) 1750130, Canadian Journal of Physics Volume 96, Number 2, February 2018

After that minor improvements i can recomend paper for publication in Symmetry journal.

Author Response

To start, I appreciate that Reviewer # 1 has recommended publication of this paper in Symmetry following minor improvements.

I fully agree with and have adopted the suggestion made by this reviewer, to extend the discussion part of the paper and compare or discuss how this approach could influence other recent results from researchers seeking to tackle these same scientific problems.  Specifically, I have written and added an entirely new section 15 titled “What is a baryon, and who ordered that?”  This is intended to be a bridge for other researchers, to help them understand their own undertakings in view of what is found in my paper, and for them to see how this paper might aid them to simplify their own work and thinking about these problems they are pursuing.  The previous section 15 is now section 16.

This new section 15 also adds a citation to one of the two papers (Canadian Journal of Physics Volume 96, Number 2, February 2018) mentioned by the reviewer, as well as to other recent papers in the same field.

The second paper (International Journal of Geometric Methods in Modern Physics Vol. 14, No. 9 (2017) 1750130 is actually more pertinent to my manuscript presently under MDPI review as entropy-996607, titled “On the thermodynamic origin of the uncertainty principle,” so I have not cited it in the current paper.

I have also made relatively minor edits throughout the balance of the paper, just for general improvement.

Reviewer 2 Report

The author discusses magnetic monopoles of SU(N) Yang-Mills gauge theory dynamics. He identifies these monopoles with baryons, fermions (inserted into monopoles) with colored quarks, and gauge boson (inside the monopoles) with gluons. The manuscript is written in a picturesque language and uses historic citations. I am not sure that the author answers Rabi’s question as he claims, but the manuscript deserves some attention.

To summarize, I recommend its publication in your journal.

Author Response

To start, I appreciate that Reviewer # 2 has recommended publication of this paper in Symmetry.

I have taken the written comments from this reviewer as suggesting improvements in three areas:  First, to remove the “picturesque” language in favor of less-picturesque descriptions.  Mainly, I suspect this has to with my language about “escorting” fermions to the monopole “dance” and the like, and so have replaced this with material having a drier tone.  Second, to go beyond the “historic” citations and provide more citations to present research by others, which overlaps with the suggestion from the first reviewer to review how my approach could influence other recent papers.  As noted in my reply to the first reviewer, this is one of the purposes of the new Section 15 titled “What is a baryon, and who ordered that?” added to the present revision of the draft manuscript.  The previous section 15 is now section 16.

Finally, the reviewer makes the fair critique that he or she remains unconvinced about whether I have answered Rabi’s question.  In the original draft I provided an outline summary in both the introduction and conclusion of why I conclude that this paper does answer Rabi’s question. To make this conclusion more widely appreciated by the readers of this journal, I have used the new section 15 to elaborate in very substantial detail how this paper does indeed answer Rabi’s question, with the hope that this enhanced detail will be more firmly convincing.

I have also made relatively minor edits throughout the balance of the paper, simply for general improvement.